# Microbes contribute to setting the ocean carbon flux by altering the fate of sinking particulates

Trang T. H. Nguyen [1], Emily J. Zakem[1], Ali Ebrahimi [2], Julia Schwartzman [2], Tolga Caglar[3], Kapil Amarnath [3], Uria Alcolombri [4], François J. Peaudecerf [4], Terence Hwa [3], Roman Stocker [4], Otto X. Cordero [2] & Naomi M. Levine [1]✉

Sinking particulate organic carbon out of the surface ocean sequesters carbon on decadal to millennial timescales. Predicting the particulate carbon flux is therefore critical for understanding both global carbon cycling and the future climate. Microbes play a crucial role in particulate organic carbon degradation, but the impact of depth-dependent microbial dynamics on ocean-scale particulate carbon fluxes is poorly understood. Here we scale-up essential features of particle-associated microbial dynamics to understand the large-scale vertical carbon flux in the ocean. Our model provides mechanistic insight into the microbial contribution to the particulate organic carbon flux profile. We show that the enhanced transfer of carbon to depth can result from populations struggling to establish colonies on sinking particles due to diffusive nutrient loss, cell detachment, and mortality. These dynamics are controlled by the interaction between multiple biotic and abiotic factors. Accurately capturing particle-microbe interactions is essential for predicting variability in large-scale carbon cycling.

[1] Department of Biological Sciences, University of Southern California, Los Angeles, CA 90089, USA. [2] Ralph M. Parsons Laboratory for Environmental Science and Engineering, Department of Civil and Environmental Engineering, Massachusetts Institute of Technology, Cambridge, MA 02139, USA. [3] Department of Physics, University of California at San Diego, La Jolla, CA 92093, USA. [4] Institute of Environmental Engineering, Department of Civil, Environmental and Geomatic Engineering, ETH Zurich, 8093 Zurich, Switzerland. ✉email: n.levine@usc.edu

The vertical flux of particulate organic carbon (POC) in the ocean has been the subject of numerous field campaigns, laboratory experiments, and modeling studies over the past four decades[e.g.,1–25]. Ultimately, the vertical flux of POC is determined by the rate of POC production in the surface ocean where de novo particle production takes place, the sinking speed of particles, and the rate of POC consumption in the subsurface ocean. Previous carbon cycle modeling studies aimed at understanding basin-scale fluxes out of the upper ocean have focused primarily on physical and chemical processes[5,7,8,26] and have shown that differences in particle size spectra[5,9,16,17], lability[6], and temperature[8,27] play important roles in determining the efficiency of POC transfer, consistent with observations[4,28,29]. However, these large-scale models have not explicitly included microbial ecological dynamics. This is problematic as many of the aforementioned physical and chemical mechanisms shown to impact the vertical flux of POC are controlled by microbial and grazer dynamics[22–25,30]. For example, lability as represented by these models is inherently related to microbial activity (as we demonstrate below). Even the size of particles and thus their sinking speed is strongly influenced by the rate of degradation by microbes and consumption by zooplankton[31–33].

Organisms alter the POC flux through a variety of processes. Heterotrophic microbes directly consume organic carbon within particles with an estimated contribution of 70-92% of POC remineralization[25]. Grazers such as zooplankton contribute to the aggregation, disaggregation, and consumption of particles[23,25,30,34–40] with small particles (0.7-53 $\mu$m) acting as a primary food source for zooplankton in the mesopelagic ocean through flux feeding[36]. However, rather than explicitly capturing these dynamics, current biogeochemical models rely on low and invariant rates of POC consumption (typically using bulk remineralization terms) tuned in order to match observed POC flux profiles[5,6,8]. Here we focus on the mechanisms behind the persistence of particulate organic carbon in the deep ocean and the role that heterotrophic microbes play in decreasing the vertical flux of particulate carbon out of the surface ocean. Additional work is needed to understand the coupled impact of microbial and grazer dynamics on the POC flux and the relative contribution of these processes to the shape of the POC flux profile (see Discussion below).

Particle-associated heterotrophic microbial communities are incredibly dynamic[41–44] and able to achieve relatively rapid growth rates (order of 1 to 10 day$^{-1}$)[32,42,43,45,46] even on seemingly 'recalcitrant' organic carbon compounds[41,44]. As a result, microbes are capable of consuming particles on the timescales of days[32,41–44]. This contrasts with observations of 'labile' particles in the bathypelagic and epibenthic zones[21,47]. Previous work suggests that temperature and pressure limitation on microbial activity might allow for the persistence of particles in the deep ocean[16,48–50]. In addition, microbial dynamics such as enzyme production, attachment, detachment, and mortality have been shown to play a key role in the rate of POC degradation[31,41–43,46,51–53]. Laboratory studies suggests that, for successful colonization of a particle to occur, particle-associated microbes must surpass a critical population size[46,54]. This critical population size or density of cells (cells per surface area) is necessary for countering the diffusive losses of both extracellular enzymes that are used to break down polymers (the main component of POC) and the resulting low-molecular-weight degradation products[42,46,54–56]. These micro-scale observations suggest that the rate of POC consumption in the ocean is highly variable and can vary as a function of microbial processes, in contrast to the conventional representation in carbon cycle models[45,46,53,57,58]. Therefore, to mechanistically understand the vertical flux of carbon in the ocean and generate robust predictions of future changes, we must account for particle-associated microbial

behavior[22,24,59], in addition to other dynamics such as particle size distribution and zooplankton activity.

Here we present a water-column model that explicitly accounts for micro-scale observations, reconciles rapid microbial growth rates with slow POC remineralization timescales (order of $10^{-3}$ to $10^{-1}$ day$^{-1}$) in the upper ocean (<2,500 m)[3,4,29,60], and determines the impact of shifts in microbial dynamics on the rate of POC flux attenuation. Specifically, we identify key aspects of particle-associated microbial community dynamics that contribute to setting the shape of the POC flux attenuation profile. We also predict that changes in microbial community dynamics can rapidly shift rates of POC remineralization. This work demonstrates that the assumption of low constant POC consumption rates is incorrect[5,61,62], and that microbial dynamics alone can generate significant variability in the POC flux. Our results challenge the classic idea of particles being inherently labile or recalcitrant and propose that lability is an emergent ecosystem property and a function of the microbial community, organic matter chemistry, and environmental conditions.

One of the difficulties in studying the vertical POC flux in the ocean is that POC flux observations provide a poor constraint on models[13]—in fact only 2 free parameters are needed to represent the classic POC flux curve while mechanistic carbon-cycle models rely on many more parameters. Thus, capturing observed POC flux profiles is not in of itself sufficient validation of a proposed mechanism. Our model allows for targeted hypotheses that can be tested in the field—the results of which will provide enhanced constraints on global carbon-cycle models allowing for more robust predictions of future changes in carbon export out of the surface ocean.

## Results

**Scaling-up micro-scale dynamics to the water-column.** Our mechanistic model captures the colonization of particles by free-living microbes and conversion from polymeric to low-molecular-weight organic matter (LMWOM) compounds as the particles sink (Methods and Supplementary Material 1). To represent the complexity of the interactions between diverse particle-associated microbial communities and a chemically diverse organic carbon pool, we use lability as an ecosystem property that specifies the conversion rate of polymeric material to LMWOM of a specific POC pool by a specific microbial group[63]. As lability is poorly constrained by observations, we test a wide range of POC lability values assuming a log-normal distribution[63], though the modeled POC fluxes at the ocean-scale are found to be independent of the specific form of the distribution chosen (Supplementary Fig. 1). We represent the organic carbon content of each model particle using a single, stochastically assigned, lability value and represent the particle-associated microbial community using a single microbial group per particle (in all we resolve 18 microbial types with different enzyme kinetics and growth rates). Sensitivity tests with complex particles, multiple microbial groups per particle, and greater numbers of microbial groups show that these simplifications do not impact the overall POC flux profiles (Supplementary Figs. 2 and 3). The model is initialized using a particle export depth between 50 and 100 m (with a default value of 100 m) with particle size distributions spanning the observed range (power law with exponent $s = -2$, $-3$, or $-4$; Supplementary Table 1). For simplicity, we do not include any additional particle formation below the export depth.

Each particle is stochastically assigned an initial radius at the export depth and type of microbial degrader. The microbial groups are defined by their enzyme kinetics (i.e., POC degradation rate),

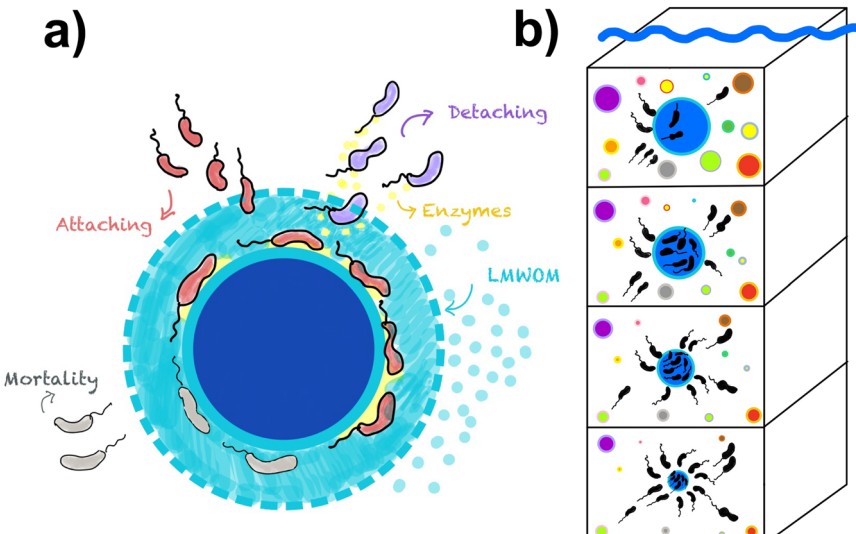

**Fig. 1 Micro-scale model dynamics. a** Illustration showing micro-scale model dynamics occurring on sinking particles. Primary degraders (red microbes) convert polymeric organic matter (dark blue sphere) into low molecular weight organic matter (LMWOM, light blue) using extracellular enzymes (yellow). The particle-associated community experiences loss due to mortality (gray microbes) and detachment (purple microbes). **b** Illustration of water column model dynamics with an emphasis on a single particle (blue sphere) falling through the water column. Each particle is stochastically assigned an initial radius, lability, and set of biological parameter values at the depth of formation (see Methods). The particle-associated microbial dynamics then evolve prognostically for each particle as it sinks through the water column and is consumed by microbial activity. The total particulate organic carbon flux throughout the water column is obtained by summing across all sinking particles.

maximum growth rate, and abundance in the free-living microbial pool, which is set to decrease exponentially with depth[64] (Fig. 1, see Methods and Supplementary Material 1 for more details). The particle is initialized with a stochastically assigned initial microbial biomass and is continually colonized by the free-living microbial pool as it sinks through the water column. The sinking speed of each particle is calculated based on its size and specific gravity[5,16,28,65] (Supplementary Material 1 eqs. s13–s15). As each particle sinks through the water column, microbial growth on the particle evolves prognostically, and organic carbon is consumed or lost to the surrounding water column due to diffusion and advection[55]. The microbially-catalyzed degradation of the particle causes the radius to shrink and the sinking speed to decrease. Temperature dependence of microbial growth rates is applied using a typical water column temperature profile[66] (Supplementary Figs. 4 and 5).

The model captures the observed density-dependent growth of particle-associated microbial populations, where the population density (cells per surface area) must surpass a critical threshold in order for successful particle colonization to occur[46] (Fig. 2). This critical threshold is a consequence of mortality and population-size dependent per capita growth rate (the 'Allee effect')[54]; see Supplementary Figs. 6–9, Supplementary Material 3. Specifically, microbial populations must overcome multiple sources of loss including mortality (e.g., viral lysis or bacterivory), detachment, and the diffusive and advective loss of LMWOM away from the particle surface (hereafter referred to collectively as loss processes). Model populations below a critical threshold cannot establish a colony on the particle due to these loss processes, consistent with laboratory observations[46] (Fig. 2 and Supplementary Fig. 10). Using a simplified mathematical model of POC degradation, we show that the population-dependent growth rate arises naturally as an interplay of simple microbial dynamics (e.g., saturating (Monod) growth kinetics, uptake kinetics, and yield) and particle chemical and physical properties (e.g., particle size and monomer diffusivity leading to nutrient loss) (Supplementary Material 4, Supplementary Fig. 11).

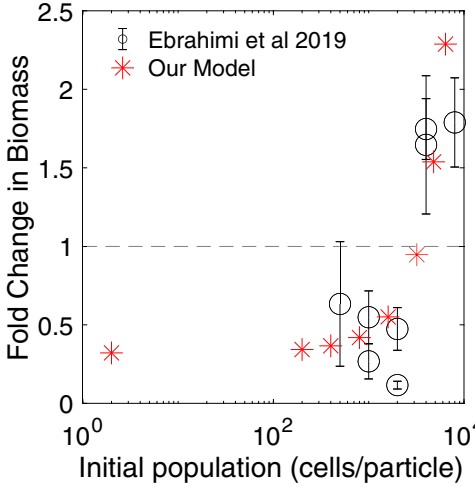

**Fig. 2 Density-dependent growth validation.** The model captures observed density-dependent growth of particle-associated communities. Fold change in microbial population biomass after 10 hours of growth is shown for the model simulations (red asterisks) and experimental data (open circles, Ebrahimi et al. (2019)). Error bars on the open circles represent standard error of measurement from at least three measurements of particle colonization density from Ebrahimi et al. (2019).

**Role of microbes in setting water-column POC fluxes.** Multiple biotic and abiotic factors control the rate of particle-associated microbial growth and therefore the microbial consumption of POC. Critically, the emergent particle consumption rates that result from interactions between these factors (e.g., particle lability and particle-microbe encounter rate) are not predictable from one factor alone (Fig. 3b). For example, with an initial radius of 500 μm and lability of 32 mmol $C_{POC}$ mmol $C_{cell}^{-1}$ day$^{-1}$, half of the time the particle is consumed in the upper 1,000 m (50% of particles) and half of the time the particle will persist into the

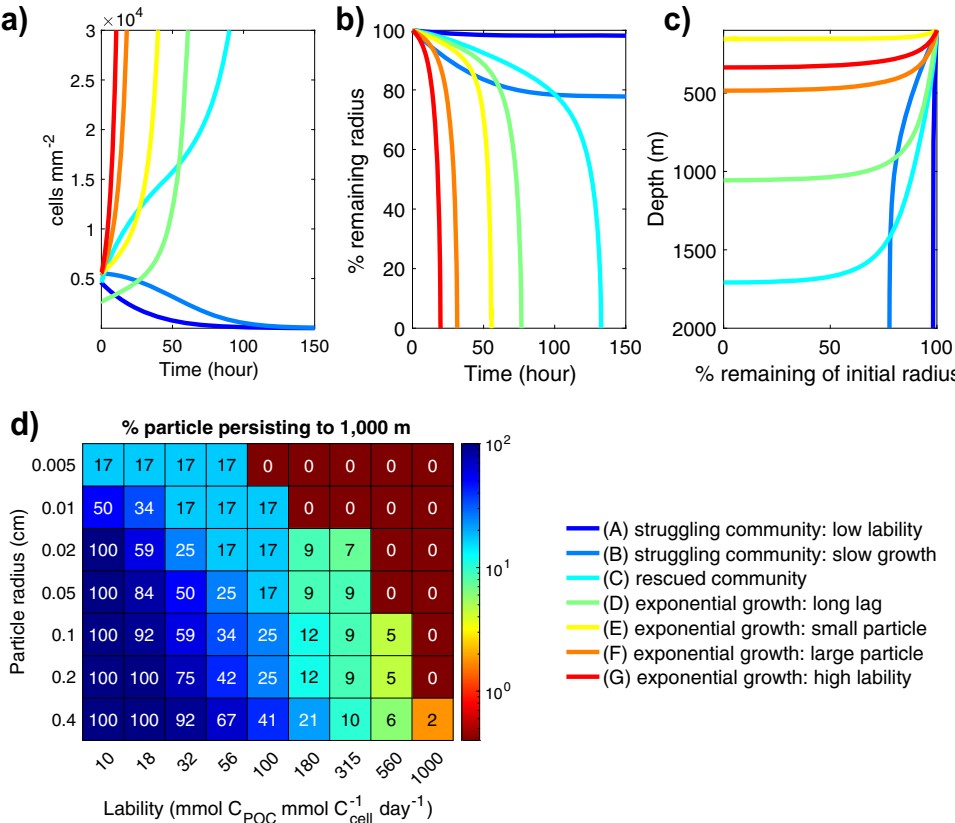

**Fig. 3 Particle-associated microbial growth dynamics.** Example degradation dynamics are shown for rapidly growing particle-associated microbial populations (curves E, F, G), 'rescued' populations (C, D), and 'struggling' populations (A, B). Parameter values are given in Supplementary Fig. 14. For each simulation, the change in microbial density on the particle surface over time (**a**), percentage changes in the particle radius over time (**b**) and depth (**c**) are shown. To quantify the impact of microbial growth dynamics on POC persistence at depth, the fraction of particles within a given lability class and radius at formation that persist below 1,000 m is calculated (**d**). The colors and values in each grid correspond to the percentage of model parameter combinations for a given lability and initial radius ($n = 72$) that persist past 1,000 m depth (see Supplementary Material 1).

deep ocean (>2,000 m). The timescale for particle degradation by the microbial community varies from ~5 days to >200 days, with most particles lasting for a few weeks below the export depth (~30 days, lognormally distributed), consistent with observations[67,68]. The degradation timescale for each particle is determined by the duration of the lag phase for the particle-associated microbial populations: once populations reach exponential growth, the particle is consumed rapidly. When population density at particle formation (100 m in the default simulations) is high, encounter rates are high, and/or lability is high, the population quickly overcomes loss processes and reaches exponential growth (Fig. 3, E-G curves). This results in the complete consumption of particles within the upper 500 m of the water column. When the population density at formation is low, encounter rates are low, and/or the lability of the particle is low, loss rates can exceed the particle-associated microbial growth rates. This results in a particle-associated community that is unable to successfully colonize the particle and reach exponential growth (here termed a "struggling" community). This in turn results in slow microbially mediated POC remineralization rates, and a higher transfer of carbon into the deep ocean (Fig. 3, A-B curves). A struggling particle-associated population can be "rescued" by recruitment from the free-living microbial pool as the particle sinks if the encounter rates are sufficiently high, resulting in shallower particle consumption (Fig. 3, e.g., A vs. C curve). The depth over which a particle sinks before it is completely consumed by the particle-associated microbial community thus depends on multiple biotic and abiotic factors that determine

microbial population behavior (Fig. 3d, see Supplementary Material 3): the conversion rate of POC to LMWOM, particle size (which determines diffusive loss and sinking speed), microbial biomass at particle formation, temperature, and exchange between the particle and free-living community (encounter rate).

The dichotomy between particle-associated populations that successfully colonize particles and populations that struggle to colonize particles provides a mechanistic explanation for the commonly used approximation of two labilities of POC[6] or the double exponential representation of the POC flux[10]. Here we show that this dichotomy emerges mechanistically as a result of microbial dynamics. Furthermore, lability cannot be defined solely as a chemical property of the particle but must be considered an ecosystem property[63,69-71] defined by both the organic carbon composition and the enzyme systems of the colonizing microbial community. The result of these dynamics is that particle remineralization rates for a single particle type (e.g., radius and lability) can vary by several orders of magnitude across particles with different microbial dynamics and over depth (Supplementary Figs. 12 and 13). This dichotomy also emerges when a modified version of the model is used where the particle-associated populations are able to directly consume POC and so are not subjected to diffusive loss (Supplementary Material 3 and Supplementary Fig. 15). In this direct-uptake formulation, the rate of particle degradation is limited by the rate at which the microbes can incorporate carbon into their biomass. We demonstrate that both models produce similar dynamics and rates of particle remineralization (Supplementary Fig. 16). Our

model mechanistically captures these particle-associated microbial dynamics and their impacts on particle consumption. This provides insight into how micro-scale processes might drive geographic variations in POC flux attenuation and allows for the generation of testable hypotheses for experimentalists.

The model predicts a shift in particle-associated microbial community composition with depth. Fast growing microbes associated with high-lability particles are abundant in the upper water column, whereas slower-growing microbes associated with low-lability particles are relatively more abundant at depth (Supplementary Fig. 17). This pattern emerges because populations growing on labile particles and populations with faster maximum growth rates can more easily surpass the critical threshold necessary for successful colonization and so thrive in the upper water column. In contrast, slower-growing communities take longer to build up biomass on particles and so are more abundant at depth. Since slow growing populations are also present in the surface ocean, the model predicts a more diverse particle-associated microbial community in the surface waters compared to the deeper ocean (Supplementary Fig. 17). These findings are consistent with documented changes in particle-associated prokaryotic communities with depth[18,57,58,72,73]: particle associated communities tend to be more homogenous at depth and community richness is positively correlated with the rate of POC remineralization.

The ocean is associated with strong vertical gradients in temperature which have been hypothesized to play an important role in POC flux attenuation[8,60,74]. We investigate the role that temperature plays in our modeled microbial dynamics using a suite of model simulations with varied temperature profiles and two temperature limitation functions, which span the observed temperature-growth rate relationship for marine microbial heterotrophs (Supplementary Fig. 5). Temperature does play an important role in the loss processes described above as decreased growth rates due to temperature limitation makes it more difficult for microbial populations to overcome loss processes and successfully colonize particles. However, our results suggest that temperature limitation is not the primary driver of the observed dynamics (Supplementary Material 3, Supplementary Figs. 18–20). Specifically, if a population is able to grow exponentially, the impact of temperature becomes secondary. Temperature becomes important for the struggling communities which are associated with slow microbial growth rates and low particle consumption rates, consistent with previous work[48,50]. The global relationships observed between POC flux attenuation rates and temperature may be partially explained by co-varying factors such as shifts in POC lability and microbial community function[4,64,75,76].

**Variability in microbially mediated POC flux attenuation.** The explicit representation of micro-scale dynamics on particles in our model generates water-column-scale estimates of bacterially mediated POC fluxes (Fig. 4). To compare across large numbers of model simulations and with observed POC flux profiles, we quantify the rate of POC flux attenuation using the commonly used power law function $F = F_{100}\left(\frac{z}{100}\right)^{-b}$, where $b$ is the attenuation exponent, which is inversely related to the vertical carbon transfer rate, $F_{100}$ is the flux at 100 m, and $z$ is depth in meters[1–3]. The model reproduces known relationships. For example, higher $b$ values in the model output are associated with higher abundance of small particles (flatter particle size spectra) and higher lability. Thus, the model predicts that oligotrophic regions with flatter particle size spectra would coincide with higher b values, consistent with previous studies[5–9,16] In addition, the model results suggest that shifts in the depth of particle formation within the euphotic zone play

a key role in the attenuation rates observed in the upper water column (<500 m) (Supplementary Fig. 21).

The observed range in the POC flux across different oceanic regions is large (i.e., 1–30% of the initial flux remains at 1,000 m)[1,3]. We demonstrate that variable microbial dynamics is sufficient to generate the observed mean, range, and distribution of observed POC flux profiles from across the global oceans ($n = 897$)[3] (Fig. 4). While other processes not included in the model (e.g., zooplankton dynamics) are also critical for setting the POC flux attenuation profile, our results suggest that variations in microbial POC consumption rates may play a significant role in determining spatial and temporal changes in the POC flux profile. For example, we show that particle-associated microbial community dynamics such as shifts in maximum growth rate or particle lability can alter the POC flux to the same extent as a change in particle size spectra (Fig. 4). These model dynamics (from rapidly growing populations to those struggling to survive) emerge as a result of stochastic interactions between biological, chemical, and physical controls on microbial growth.

**Discussion**

Our results suggest that the stochastic assembly of communities on particles frequently results in communities struggling to overcome losses, thereby generating the long tail of persistent POC at depth (>1000 m). Furthermore, for particles that persist at depth due to sub-optimal growth of the particle-associated communities, relatively small changes in microbial dynamics can rescue these communities, allowing them to rapidly reach exponential growth and generating large changes in POC flux. For example, while a given particle type (e.g., size and lability) might persist below 1000 m in one region of the ocean due to low initial biomass or low microbial encounter rates, that same particle might be consumed rapidly in a different region where microbial encounter rates are higher (Fig. 3). Our model provides an alternative hypothesis for pulsed export of carbon from oligotrophic regions[77–79]. In addition to a shift in particle size associated with these export events[e.g.,77,80,81], the microbial community in oligotrophic regions is typically associated with picoplankton dominated communities and so may not be adapted to consume particles generated by larger phytoplankton groups. These differences in microbial dynamics, represented as lower biomass at formation and lower encounter rates in our model, will yield greater transfer efficiency of carbon to depth than would occur if the same type and quantity of particles were released in more productive regions where the microbial community is primed to consume POC generated by larger phytoplankton groups.

This work suggests that shifts in microbial communities both in surface waters and at depth can result in significantly different POC fluxes and that the magnitude of microbial driven variations in the POC flux is similar to other previously proposed processes (e.g., particle size spectrum). This is not to say that other processes do not also impact the POC flux. Dynamics not explicitly represented in the model such as particle aggregation and disaggregation[32,35,39], zooplankton grazing on particles[30,36,37,40,82], phytoplankton dynamics[83–85], and the formation of new particles within the water column[83–85] also play an important role. An exciting avenue of future work is to investigate the extent to which complex ecological interactions between the microbial communities and zooplankton dynamics impact the POC flux, the relative importance of these different processes for determining the rate of organic carbon export, and how these dynamics may vary geographically. Our model provides a unique framework for understanding the carbon cycle impact of zooplankton bactivory on particle-associated communities[42], and the direct consumption of sinking particles

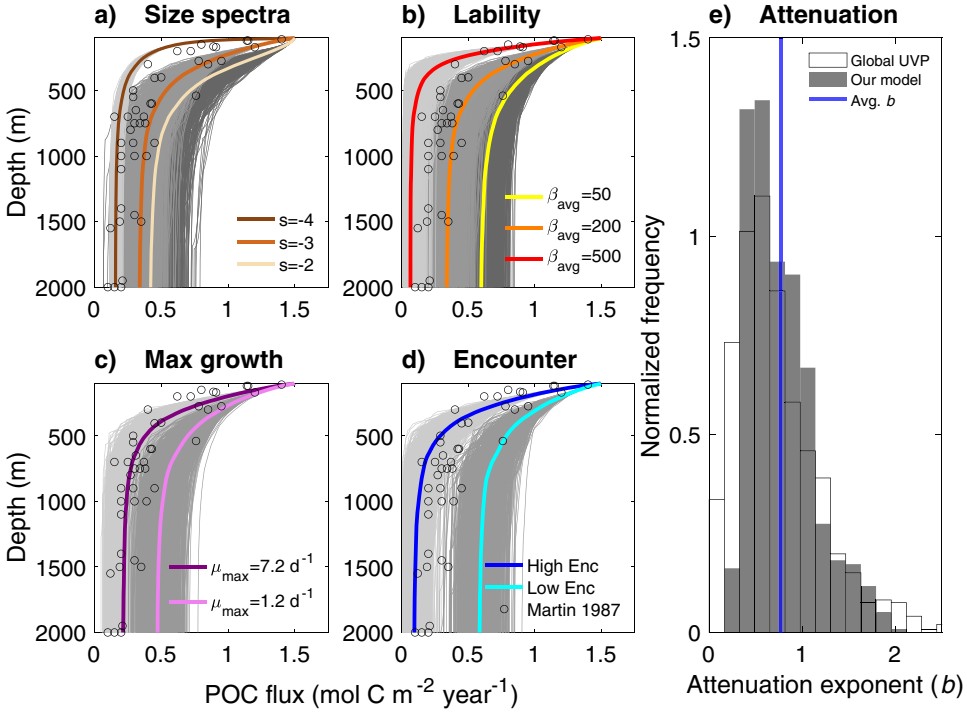

**Fig. 4 Microbial contribution to large-scale particulate organic carbon (POC) fluxes.** Shifts in POC fluxes over depth are shown as a function of varying particle and microbial dynamics. Each gray line is the integrated flux over 2,256 to 763,615 stochastically initialized particles (depending on the particle size spectra). For each parameter set, 1,000 stochastic simulations were conducted (1,000 gray lines of different shades). The average for each parameter set is shown with the thick colored line. The data from Martin *et al* (1987) (in open circles) are also shown. POC transfer efficiency decreases (larger attenuation exponent $b$) with more negative particle size spectra power law exponent $s$ (**a**), higher mean particle lability ($\beta_{avg}$, mmol $C_{poc}$ mmol $C_{cell}^{-1}$ day$^{-1}$) (**b**), higher maximal growth rate ($\mu_{max}$, day$^{-1}$) (**c**), and higher particle-microbe encounter rate (Enc) (**d**). The distribution of attenuation exponent ($b$) for all 10,000 stochastic simulations from panels a-d (gray bars) are compared against observed attenuation coefficients from 897 global Underwater Video Profiler (UVP) measurements compiled by Guidi et al. (2015) (open bars) (**e**).

by zooplankton (e.g., exploring preferences for different types of POC and particle sizes). Direct inclusion of dynamics such as zooplankton disaggregation of particles and zooplankton vertical migration would allow for further mechanistic constraints on spatial and temporal variability of organic carbon export.

This work takes a first step towards explicitly integrating micro-scale dynamics into large-scale models to generate predictions of organic carbon fluxes in the ocean. We show that microbial growth dynamics can generate temporal and spatial variability in POC consumption rates, suggesting that current parametrizations for POC degradation are inadequate (Supplementary Material 1–3). Our model generates hypotheses as to the relative importance of particle-associated microbial dynamics throughout the water column that can be tested by targeted field and laboratory studies. These studies will in turn improve the model parameterizations and generate more robust estimates of the POC vertical flux. For example, incubation experiments could test whether a significant fraction of particle-associated populations at depth are below the critical density threshold. Many aspects of particle-associated microbial dynamics are currently poorly constrained (e.g., encounter rates, bacterial growth efficiencies). Our results highlight the need for better in situ measurements of these key biological processes such as loss processes for microbial communities, microbial abundance on particles, enzyme activities, growth rates on particles, and encounter and detachment rates for dominant particle-associated marine species. Ultimately, robust predictions of future shifts in carbon cycle dynamics require accurate, mechanistic representation of the primary processes in global climate models. Here we demonstrate that particle-associated microbial dynamics are one of these processes.

## Methods

This model captures key micro-scale dynamics occurring on particulate organic carbon (POC) in a manner scalable to the water column. For the results presented in the main text, we represent particle-associated microbial diversity using 18 groups of heterotrophic microbes, defined based on their enzyme kinetics and growth rates. Supplementary Material 2 presents a sensitivity test with a continuum of microbial classes and demonstrates that this discretization does not impact our results. We also make the simplifying assumption that each particle type consists of a single lability of organic carbon colonized by a single type of microbial primary degrader, though the conclusions of this work are not dependent on this assumption (Supplementary Fig. 2 and Supplementary Material 2). Here we track each unique particle type $i$, which is defined based on radius at formation and lability ($\beta_i$). We include enzymatic degradation of polymer into low molecular weight organic matter ($C_{lmwom}$), density dependent growth of the particle-associated microbial community ($B_i$), and the attachment ($E_{i,z}$) and detachment ($L_i$) of heterotrophic microbes to/from the particles. This model can be coupled to a full ecosystem model such that the generation of each particle type can be calculated prognostically. However, here we focus on the degradation of POC below the export depth and so simply include a source term to represent net particle formation above the export depth (default 100 m, see Supplementary Material 2 for simulations with alternative formation depths). For simplicity, we also do not allow for aggregation or disaggregation to occur within the water column. An extended model description is provided in Supplementary Material 1.

The change in the carbon content of particle $i$ ($C_{poc,i}$, mmol $C_{POC}$ particle$^{-1}$) over time is defined as:

$$\frac{dC_{poc,i}}{dt} = -\beta_i B_i \tag{1}$$

where $\beta_i$ (mmol $C_{POC}$ mmol $C_{cell}^{-1}$ day$^{-1}$) represents the polymer degradation rate of $C_{poc,i}$ by microbial group $B_i$ (mmol $C_{cell}$ particle$^{-1}$) similar to[59]. Specifically, $\beta_i$ captures differences in 'lability' of particles, which is a function of the organic carbon itself, the microbial enzymes specific to group $B_i$, and production rate of those enzymes by $B_i$. When the particle is fully consumed ($C_{poc}=0$), the particle-associated microbial community detaches and so consumption stops.

The enzymatic degradation of POC results in the production of low molecular weight organic matter (LMWOM) ($C_{lmwom}$, mmol C m$^{-3}$ particle$^{-1}$) which supports microbial growth. We assume that there is no loss of carbon during the enzymatic cleavage from POC to LMWOM such that 1 mmol $C_{poc}$ degraded = 1

mmol $C_{lmwom}$ produced. There is however diffusive loss of LMWOM away from the particle as described in Eq. 2. Specifically, the LMWOM concentration is calculated assuming steady state dynamics as:

$$C_{lmwom} = \left( \beta_i B_i - \frac{\mu_i B_i}{y_{lmwom}} \right) / d_{loss} \qquad (2)$$

where $\mu_i$ is the growth rate of microbial group $B_i$ on the particle (day$^{-1}$), $y_{lmwom}$ is the aerobic microbial growth efficiency (mmol $C_{cell}$ mmol $C_{lmwom}$ $^{-1}$), and $d_{loss}$ is the diffusion loss rate of the LMWOM (m$^3$ day$^{-1}$) (Supplementary Material 1 eq. s3 and Supplementary Material 4 for full calculation).

Microbial dynamics on each particle are defined as:

$$\frac{dB_i}{dt} = \left( \mu_i - L_i - m_{lin,i} \right) B_i + E_{i,z} \qquad (3)$$

where $m_{lin,i}$ is the microbial mortality rate (day$^{-1}$) and $L_i$ is the detachment rate (day$^{-1}$). The microbial encounter rate ($E_{i,z}$, mmol $C_{cell}$ day$^{-1}$ particle$^{-1}$) represents the rate of colonization of the particle by the free-living microbial pool. $E_{i,z}$ varies with depth based on particle size, sinking speed, and the abundance of free-living, motile microbes in group $i$[51,64] (eq. s11 in Supplementary Material 1). The free-living abundance is assumed to decrease exponentially with depth[64]. Microbial growth rate, $\mu_i$ (day$^{-1}$) is dependent on the LMWOM concentration at the particle surface and is represented with the Monod equation:

$$\mu_i = V_{max,i} \frac{C_{lmwom}}{C_{lmwom} + k_{m,i}} \gamma_{T,z} \qquad (4)$$

where $V_{max,i}$, $k_{m,i}$, and $\gamma_{T,z}$ represent the maximum LMWOM uptake rate, the half saturation of LMWOM uptake by microbial group $B_i$, and the temperature limitation at depth z (eq. s11 in Supplementary Material 1), respectively. Model parameter values are given in Supplementary Table 1 and sensitivity tests are described in Supplementary Material 2.

The total POC flux at a certain depth z ($F_{poc,z}$ in mmol $C_{POC}$ m$^{-2}$ day$^{-1}$), is calculated as the sum of the vertical fluxes of each individual particle as they sink through the water column, where $F_{poc,z}$ is:

$$F_{poc,z} = \sum C_{poc,i,z} N_{i,z} \omega_{i,z} \qquad (5)$$

where $N_{i,z}$ is the number of particles of type $i$ per m$^3$ water column at depth z.

To test the impact of particle-associated microbial dynamics on the POC vertical flux through the water column, we perform a set of stochastic simulations in which parameters are randomly chosen from within a reasonable range. Here we simulate a single 2.24 m × 2.24 m water column initialized between 50 and 100 m with an observed particle size distribution yielding a total flux of 1.5 mol m$^{-2}$ yr$^{-1}$ [1,5,13]. Specifically, we simulate 2,256 particles (particle size spectra with power law exponent $s = -2$), 69,000 particles ($s = -3$), or 763,615 particles ($s = -4$) ranging from 50 $\mu$m to 0.4 cm in radius as they fall through the water column. For each particle, the following model parameters are stochastically assigned within a reasonable range (Supplementary Table 1) using a uniform distribution, except for lability for which a log-normal distribution is used: maximum growth rate $V_{max}$ (1.2 or 7.2 day$^{-1}$), particle lability $\beta$ (10–1,000 mmol $C_{POC}$ mmol $C_{cell}$ $^{-1}$ day$^{-1}$), initial cell density (400–2,800 cell mm$^{-2}$), and density of free-living community of microbes in group $i$ ($F_i$) (10–285 cell mm$^{-3}$). The stochastic simulations are conducted 1,000 times for each particle size distribution. Simulations are run for 600 days which is sufficient for all particles to be fully consumed or exported to >4,000 m. The attenuation exponent $b$ for the modeled POC flux is calculated using a least square fit of the power law function.

**Reporting summary**. Further information on research design is available in the Nature Research Reporting Summary linked to this article.

## Data availability
The microbial growth data generated in this study are provided in the Supplementary Table S2.

## Code availability
The model code is deposited on GitHub (https://github.com/LevineLab/POMmodel) and citable using https://doi.org/10.5281/zenodo.6015020.

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

## Acknowledgements

This work was supported by a grant from the Simons Foundation (542387 to NML, 542395 to RS and OC, 542389 to TH). We thank N. Norris and E. Lee for assistance with model development.

## Author contributions

TTHN and NML designed the study, developed the full model, and conducted the numerical analysis; TC and TH developed and analyzed the mathematical model in Supplementary Material 4; KA and TC performed measurements of bacterial parameters for *Vibrio* sp. 1A01; EJZ, OXC, AE, JS, UA, FJP, TH, and RS contributed data and to model development; all authors contributed to the writing.

## Competing interests

The authors declare no competing interests.
