## [Peer Review File · Nature Communications]

Title: Microbes contribute to setting the ocean carbon flux by altering the fate of sinking particulatesREVIEWER COMMENTS

Reviewer #1 (Remarks to the Author):

Dear Editor and Authors,

Thank you for this opportunity to review the manuscript “The effect of microbial activities on ocean-scale carbon transport” by Nguyen et al., submitted to Nature Communications.

This manuscript provides a model based characterization of microscale microbial dynamics on different kinds of sinking particles. It examines how those dynamics affect carbon flux into the deep ocean. It is the most sophisticated model of microbial processes on particles that I have encountered. In particular it provides a first look at how particle lability and other particle properties might impact microbe-particle interactions. Nguyen and team make the case that the balance of microbial growth and loss processes lead to some particles that essentially are not remineralized by microorganisms and so contribute the most to carbon flux. In these “recalcitrant particles” microorganisms never establish significant numbers to cooperate to break down the particles.

The manuscript provides compelling evidence that microscale microbial processes on particles have the potential to dramatically impact carbon flux. It provides a valuable scaffold for future microbial-particle interaction models, and provides testable hypotheses that I suspect other teams will want to explore with experiments and field observations. Thus, I think it will be of broad interest to anyone who is interested in the marine carbon cycle, and those interested in microbial ecology.

I very much enjoyed reading this manuscript. I think it is a great fit for Nature Communications. It is thorough and concise, and very well written, to the point that I have very few specific comments on the writing. It is thorough in the sense that the authors validate many aspects of the model, and test for sensitivity of many of their model parameters. Despite this attention to detail, the authors somehow managed to fit all of this analysis and its significance into an effectively six page manuscript.

I am going to push back somewhat on one aspect of the model. From my reading, in all simulations, the authors contend that microorganisms use extracellular enzymes to break the particle down into low-molecular weight organic matter (LMWOM). The microbes then collect that LMWOM out in a diffusive boundary zone just above the particle. I have no doubt that this process accurately describes the foraging behavior of *some* particle associated organisms. As the authors point out, Ebrahimi et al. 2019 provide an elegant model showing how this cooperative process would work, and then provide experimental evidence that at least one bacterial species *Psychromonas* sp. *psych6C06*, shows this cooperative behavior when it colonizes spherical chitinous beads. However, I’m not convinced that it describes the foraging behavior of *all* particle degrading microorganisms, or even most particle degrading microbes. One could imagine microorganisms degrading polymers directly with surface attached enzymes and then transporting the LMWOM immediately across their cell membrane without giving it time to diffuse. One could also imagine that the particle's fractal structure could lead to internal pockets where a microbe might produce extracellular enzymes which would diffuse into that pocket where the microbe might consume the LMWOM, without the compounds having an opportunity to diffuse away. Similarly, particles might just leach consumable LMWOM on their own and microbes might consume those directly. In any of these situations I think the scenario where microbes go effectively extinct on some or many particles would be less likely to occur. I furthermore suspect that at least some

microorganisms might carry out some process other than cooperative degradation on most particles.

Some particles, especially the ones that are a lot like spheres of chitin, might not have many microbes that carry out cooperative processes on them, but many particles would likely still have some microbes degrading them, even if none of them were exo-enzyme producing cooperators. In the presence of non-cooperative degraders, my intuition is that you would rarely/never have local microbial extinctions on particles, per figure 1A simulations A and B. However, I think the other processes, especially variability in particle lability, would likely still have a big impact on flux.

I don't think this is a fatal flaw, since an inability of cooperative species to degrade particles could indeed be a factor that decreases the remineralization of particles (and increases DOM flux), especially if cooperative particle degradation ends up being the most efficient way to consume a particle. That said, I think either more caveats, or if I have misunderstood the situation, counter arguments, would strengthen the paper.

As a more technical consideration: I think figure 2A could perhaps be presented a little differently. I'm struggling to internalize the differences between the model runs A-G.

I get that the main point is that they have some runs where the microbes die out, some where they grow exponentially, and some where they struggle at first and then "recover". That said, I have trouble keeping track of which parameters are varying in which runs. I don't insist on this, but it would sure be nice if the authors could think about laying out the different parameter sets in a way where it is more intuitive for the reader to keep track of which things are different between which runs.

Also 2A: Why did they pick these parameter sets and not others? Is it because these give good examples of different microbial dynamics?

One more question about 2A: panel 1: In the model runs C-G where the microbes do grow, their numbers appear to approach infinity. I presume there is some limit to microbial abundance on particles in this model. Why do these not appear to have any upper asymptote?

Figure 3: The gray lines tell us about the spread of all of the runs, but they don't tell as much about the spread of the runs within the groups. I am wondering if it would be possible to add standard deviations, or color code the individual runs so that we can see the within run spread of the different sets of runs. 3D: Is this relationship between encounter rate and flux at all dependent on the initial colonization density of the particles? My hunch is that the encounter rate is likely less important if the microbes are pre-colonized.

Figure S4 -- Other publications have suggested a strong relationship between temperature and particle flux (eg DeVries and Weber 2017) The relationship between temperature and flux seems to be minimal when a Temp-Off (constant temperature) model is compared to a Temp-on one where temperature varies through the water column. What if the temperature through the water column varies as seen in different ocean regions? Maybe such analysis is beyond the scope of this paper. And indeed, it is technically true as the authors say that "Temperature is not the primary driver of the observed

dynamics" (lines 93-94), since the observed dynamics only occur with the aforementioned temperature profiles. However, it's worth noting that temperature is still likely an important driver of region-to-region particle flux, and that the authors have not shown that temperature is unimportant, just that it doesn't matter much in this model when only one temperature profile was considered.

Line 119-120 "A struggling particle-associated population can be rescued by recruitment from the free-living bacterial pool as the particle sinks..."

I see your examples of this happening (eg Figure 2 C and D), but I'm still not sure how this process works. Do encounter rates sometimes increase with depth? I could imagine a population struggling and then the particle sinks into a region with high encounter rates. And then the encounter rates exceed mortality until the population reaches the threshold where cooperation between the microbes can break down the particle. But if encounter rates scale with microbial abundances which are either most abundant at the surface, or decreasing with depth, I'd assume the attachment would either exceed mortality at the surface, in which case no "rescue" is necessary, or it wouldn't and the population would crash. What am I missing here?

Line 198-201: "We show that POM consumption is temporally and spatially variable as a function of microbial growth dynamics... Supplement S1, S2, S3" -- I'm not sure I see how they show this. Are they saying that dynamics are different from one particle to another and between depths? As written, this implies that the authors have evidence that these factors vary between different ocean regions, or perhaps seasonally, but I don't think they tested such variability.

Line 211 "single type of bacterial primary degrader" -- One way that I misread this at first was that I thought that the model assumed that there was one type of bacterial degrader *total*. But actually there is one bacterial degrader *per particle* right? And those different degraders have different growth rates and temperature sensitivities and so on. In either case, the authors might consider clarifying a bit more in the text.

Reference:

DeVries T, Weber T. The export and fate of organic matter in the ocean: New constraints from combining satellite and oceanographic tracer observations. *Global Biogeochem Cycles*. 2017 Jan 1;2016GB005551.

Signed,
Jacob Cram

Reviewer #2 (Remarks to the Author):

Review of Nguyen et al. for Nature Communications

General

Given the lack of detailed mechanistic advances in teasing apart the different facets of the oceans' biological pump this is a very timely study. It builds on conceptual papers from Aumont et al. (2017), Zakem et al. (2021) and on some new Twilight Zone omics work from Ed de Long's lab (Foff et al., 2021). It will be a useful stepping stone to improving our insights into how changing climate will influence the oceans' biological pump – as stated in their Abstract.

There was one thread throughout the manuscript that I was surprised to see – one that was centred on equating the “well-known POC flux profile” solely with microbial dynamics (rates, fluxes, community structure). This profile is the cumulative imprint of microbes, zooplankton flux feeding, and at depth (diurnal zooplankton migration) with little known about what the depth-dependency of each of these processes are, their relative contribution to flux attenuation, and the implications of discrete (flux feeding) versus (continuous) processes in setting this flux profile (see conceptual figure in Buesseler and Boyd, 2009, and also modelling studies by Jackson and Burd on flux feeding simulations).

Thus, there are issues around assuming that this flux profile is exclusively (or predominantly) set by microbes and hence the value of looking for match-ups with this flux profile is questionable. I think the authors can get around this impasse by exploring studies that report on the depth dependency of rates (rather than community structure/ 16S ...) such as for enzymatic activities, remineralisation, etc. What type of profiles do we see from such microbial processes.

Specific

Abstract

18 – crucial role – yes – but can we quantify it – given that many prior studies using imperfect tools (leucine, thymidine etc) at atmospheric pressure do a poor job of balancing the C demand vs C availability (inferred from the particle flux attenuation profiles).

19-22 – see general comments above.

23 – enhanced transfer where?

24 anthropomorphic terms – struggling !

what about bacterivory – see movies from Kiorboe lab and patterns of colonisation from this group.

Introduction line 24 – what about zooplankton flux feeding – see Jackson and Burd modelling papers.

– sinking speed and zooplankton mediated particle transformations. 40 – is it solely microbial POC consumption that drives these models – or tuning the ‘remineralisation’ as a bulk term?

‘recalcitrance is in the eye of the solubiliser’!! It's a relative term – as we know from the soil C literature.

– please cross check these papers to see if they were done at ambient pressure (sea level). See Christian Tamburini's pressure work.

– what about the role of the diffusion boundary layer in these processes?

– see general comments – this all needs careful re-consideration.

64/65 – ditto – needs much more stringent justification – see general comments

– ditto – revisit these after resolving the following from above.

“This profile is the cumulative imprint of microbes, zooplankton flux feeding, and at depth (diurnal zooplankton migration) with little known about what the depth-dependency of each of these processes are, their relative contribution to flux attenuation, and the implications of discrete (flux feeding) versus (continuous) processes in setting this flux profile.”

Results – their interpretation will need to be carefully revisited – see general comments above.

– some sensitivity analysis on using a 100 m depth is essential – given that the stratum of de novo particle synthesis can be much shallower (50 m or less and varies regionally) and that this region and the underlying 50-100 m band of waters exhibit major gradients with decreasing stocks – then the depth where microbial colonisation commence will be a key determinant of whether groups can attain critical particle population sizes. This regional variation in this depth will influence the regional model comparisons again using matchups with the POC flux profiles – which needs to be reconsidered.

– an intriguing finding – worthy of a mention in the Abstract?

– half of the time?? vague

deep ocean?? Vague

– see comment above on arbitrary 100 m initialisation depth.

struggling?

rescued?

121-124 – see general comments above.

thriving struggling??

give an example of a single particle type please

– see comment on the need for regional variations in the (100 m) initialisation depth.

to 144 – this demarcation is a ‘self-fulfilling’ outcome ?

– some of these references refer to model communities

– in particular – see general comments above!!

Discussion – will need to be overhauled in line with General comments above.

For example line 184

191

I identify myself as a reviewer – please pass on my name to the authors

Thanks

Philip Boyd

Reviewer #3 (Remarks to the Author):

The vertical flux of POC in the ocean decreases roughly exponentially with depth. This depends on a great many factors, including, for example, surface ocean production, community composition, particle characteristics, mineral ballast, the mineralization of organic matter, and everything that impacts those. Nguyen and colleagues contribute to the discussion of the processes that controls this flux by presenting a numerical model that focuses on the dynamics of heterotrophic bacteria that is responsible for POM mineralization. A strong motivation for their work is the desire to connect the observed POC fluxes with the underlying mechanisms acting at the particle level, thereby improving the ability to capture and explain the observed variability and ultimately the predictions of the future biological pump. The manuscript is well written, timely and important.

The authors demonstrate that POM mineralization rates can be high and variable due to the underlying microbial dynamics, producing results consistent with the observed flux - depth relationships. In their model, POM degradation progresses via the formation of low molecular weight (LMW) OM, which then is either consumed or lost from the particle to the surrounding water and is assumed to be at steady state. Bacterial biomass is affected by growth, with growth rates depending on growth efficiencies, and LMWOM concentrations, death, detachment and encounter rates. The model emphasis on microbial dynamics follows in the footsteps of Mislán et al. 2014, but it expands on it by looking at a number of aspects relevant for the biological pump, for example particle size distributions etc. Here, the particle radius (selected from realistic particle size distributions) sets the initial POM, which is then modified due to reaction, reducing particle diameters and sinking speed.

There are many ways to approximate each of the processes considered, but I found the formulations chosen to be reasonable and well informed by existing work. Importantly, the authors provide a detailed assessment of the impact of some of the key assumptions made when translating the complex system of sinking particles and associated microbial community into a very simple set of equations. The model is described clearly and relatively easy to follow, and it nicely includes the consideration of fractal dimensions and hydrodynamics in the parameterization. In several instances, when different parameterizations seemed reasonable or a conceptual approach gave me pause, the authors demonstrated through sensitivity tests that the impact on the outcome is small (supplement S2; examples include the particle size distribution, the one microbial community per particles etc.).

Some suggestions for potential improvements:

- organic mineralization rate: POM to LMWOM transformation is proportional to a cell specific rate β and the cell density of the heterotrophic bacteria on each particle. There is no explicit dependency on the amount of organic matter (e.g. Gloege et al. 2017 *Global Biogeochem. Cycles* 31, 1192-1215; Aumont et al. 2017, *Biogeosciences* 14, 2321-2341). This reflects a particular conceptual model on the

rate limiting steps I consider reasonable, and the impact of OM is simulated via a feedback on the bacterial cell numbers. Furthermore, the tests considering single vs. 25 lability types in a particle (S2) seem to suggest that the timescales are such that a potential POM dependency is captured through this mechanism. But have you considered POM-dependent kinetics and if so, can you confirm that the outcome is not sensitive to this selection? And in the model, do you continue to consume POM when POM is 0 but $B > 0$ (eq. 1; is B considered part of POM?) What kind of bacteria/POM ratios are you obtaining in the simulations? (Line 242: point to eq. 11, not 13; in eq. 5 consider a discrete summation)

- missing processes: Particle aggregation and disaggregation or zooplankton grazing are generally considered to be important and have improved model fits to fluxes, yet are not considered here (e.g. Karakas et al. 2009, *Progress in Oceanography* 83, 331-341; Burd and Jackson 2009 *Annu. Rev. Mar. Sci.* 1, 65-90). Zooplankton dynamics can contribute to the spatial and temporal variability in fluxes (e.g. Jackson and Burd 2002, *DSR II*, 49, 193-217; Cavan et al. 2017 *Nat. Comm* 8:14847, considering the effect of zooplankton in the context of OMZs; Boeuf et al. 2019. *PNAS* 116, 11824-32, suggesting that “a considerable amount of sinking POM degradation may be due to its consumption by mid- and deep-water animals, and their associated deep-sea adapted microbiota”), and that the formation and breaking up of aggregates are critical for the determination of sinking velocities, time for remineralization and hence C fluxes.

The nagging question is to what extent ignoring these processes yet matching the Martin curve overemphasizes the role of the processes highlighted here. To some extent, the authors address this in the Supplement S3, but a clear(er) discussion of what measurements are needed to assess the relative importance of microbial vs. the aggregation/disaggregation processes on fluxes would be helpful. Consider to explicitly identify the specific measurements your model results indicate that will distinguish the numerous processes that may explain observed fluxes (fig 3), high growth rates and variable POM consumption rates (line 66ff). This would further bolster the comments made regarding the need for better in situ measurements (paragraph line 190ff) and testable hypotheses (line 69).

**Reviewer #1 (Remarks to the Author):**

*Dear Editor and Authors,*

*Thank you for this opportunity to review the manuscript “The effect of microbial activities on ocean-scale*
*carbon transport” by Nguyen et al., submitted to Nature Communications.*

*This manuscript provides a model-based characterization of microscale microbial dynamics on different*
*kinds of sinking particles. It examines how those dynamics affect carbon flux into the deep ocean. It is the*
*most sophisticated model of microbial processes on particles that I have encountered. In particular it*
*provides a first look at how particle lability and other particle properties might impact microbe-particle*
*interactions. Nguyen and team make the case that the balance of microbial growth and loss processes*
*lead to some particles that essentially are not remineralized by microorganisms and so contribute the*
*most to carbon flux. In these “recalcitrant particles” microorganisms never establish significant numbers*
*to cooperate to break down the particles.*

*The manuscript provides compelling evidence that microscale microbial processes on particles have the*
*potential to dramatically impact carbon flux. It provides a valuable scaffold for future microbial-particle*
*interaction models and provides testable hypotheses that I suspect other teams will want to explore with*
*experiments and field observations. Thus, I think it will be of broad interest to anyone who is interested*
*in the marine carbon cycle, and those interested in microbial ecology.*

*I very much enjoyed reading this manuscript. I think it is a great fit for Nature Communications. It is*
*thorough and concise, and very well written, to the point that I have very few specific comments on the*
*writing. It is thorough in the sense that the authors validate many aspects of the model, and test for*
*sensitivity of many of their model parameters. Despite this attention to detail, the authors somehow*
*managed to fit all of this analysis and its significance into an effectively six-page manuscript.*

**We thank you for your positive and constructive feedback! We feel you have truly helped us make this a**
**stronger paper. We would like to explicitly acknowledge your contribution in the acknowledgements if**
**you are OK with this.**

*I am going to push back somewhat on one aspect of the model. From my reading, in all simulations, the*
*authors contend that microorganisms use extracellular enzymes to break the particle down into low-*
*molecular weight organic matter (LMWOM). The microbes then collect that LMWOM out in a diffusive*
*boundary zone just above the particle. I have no doubt that this process accurately describes the foraging*
*behavior of *some* particle associated organisms. As the authors point out, Ebrahimi et al. 2019 provide*
*an elegant model showing how this cooperative process would work, and then provide experimental*
*evidence that at least one bacterial species Psychromonas sp. psych6C06, shows this cooperative behavior*
*when it colonizes spherical chitinous beads. However, I’m not convinced that it describes the foraging*
*behavior of *all* particle degrading microorganisms, or even most particle degrading microbes. One*
*could imagine microorganisms degrading polymers directly with surface attached enzymes and then*
*transporting the LMWOM immediately across their cell membrane without giving it time to diffuse. One*
*could also imagine that the particle's fractal structure could lead to internal pockets where a microbe*
*might produce extracellular enzymes which would diffuse into that pocket where the microbe might*
*consume the LMWOM, without the compounds having an opportunity to diffuse away. Similarly, particles*
*might just leach consumable LMWOM on their own and microbes might consume those directly. In any*
*of these situations I think the scenario where microbes go effectively extinct on some or many particles*
*would be less likely to occur. I furthermore suspect that at least some microorganisms might carry out*
*some process other than cooperative degradation on most particles.*

We completely agree that there are other potential ways in which microbes can degrade particle as Dr.
Cram very nicely describes. Our original model formulation focused on POM degradation through
extracellular enzymatic cleavage of polymers into smaller compounds (LMWOM) as we believe this is a
dominant mode of organic matter degradation¹⁻⁶. We also agree that the concentration of LMWOM
surrounding the particle is almost certainly not homogeneous such that the areas with high densities of
microbes (that are actively degrading the polymer) will have a higher concentration than areas with fewer
degraders. These microscale gradients were represented in the Ebrahimi model (2019)⁷. We show that our
coarse-grained model captures the same overall behavior as the individual-based micro-scale model. It is
also important to note that while the model captures the cooperative degradation that Ebrahimi et al
documented, the model formulation does not explicitly represent cooperation in the sense that enzyme
production is not a function of microbial population size. We show that diffusive loss of LMWOM away
from the particle is in of itself a sufficient loss term to impact particle-associated microbial dynamics (*see*
*calculation in response below*). As a result, we believe that the current model captures the dynamics of
microbial degraders who use enzymes to extracellularly cleave polymers to LMWOM on the surface of
particles (whether or not they are working cooperatively).

However, Dr. Cram is correct that the original model does not capture either the dynamics of degraders
who are able to directly take up polymers and degrade them intercellularly (i.e., selfish-uptake⁸⁻¹⁰) or of
degraders caught within the matrix of a particle where diffusive loss would be minimized. To understand
the impact that this alternative uptake strategy might have on particle degradation and the findings of our
paper, we have developed a modified version of the model that simulates direct uptake by the degraders
(i.e., no LMWOM is generated) (described in *Supplement S3*). We ran this model using a similar range of
parameter values (initial biomass, free living concentration, particle size, etc.). Lability in this model must
be treated differently – here we modify the β term in the original model to ξ which is constrained by the
rate at which microbes can incorporate carbon into their biomass (i.e., microbes cannot degrade the particle
faster than they can grow). We demonstrate that the three types of particle degradation profiles also emerge
for direct-uptake degraders: 1) exponential growth and rapid consumption, 2) populations struggling to
overcome loss (here only mortality and detachment rate), and 3) rescued populations due to encounter
with free-living bacterial communities (as shown in *Supplement Fig. S14*). Of note, the ‘rescued’
populations are notably more common in the enzymatic cleavage model than in the direct uptake model.

This alternative mechanism for POM degradation does not change the primary finding of our works: POM
consumption is temporally and spatially variable as a function of microbial growth dynamics and
microbial community composition. Changes in microbial processes (e.g., enzyme kinetics, loss rates) have
the potential to significantly alter the rate of particle degradation and thus the POM flux. We describe this
new model in *Supplement S3*. We have added additional text (lines 155-160) and supplement figures
(*Supplement Figs. S13-15*) summarizing these new model findings. We also have added text highlighting
the need to constrain loss processes for these communities as this is the key driver of whether the
populations successfully colonize particles or struggle to survive (lines 252-254).

Finally, for this study we focus on surface associated microbes as these groups are associated with key
dynamics that we are interested in, specifically encounter and detachment. While consumption of POM
due to microbes in a particle interior undoubtedly plays a role, many of these processes are assumed to be
limited by oxygen and so occur at much slower rates than aerobic consumption on the particle surface¹¹.
Such consumption would contribute an additional loss of POM (higher degradation rate) that is not
currently accounted for in the model. However, these dynamics should be similar to the new direct uptake

model and so we would not expect this process to significantly alter the shape of the POM flux.
Investigating the relative importance of aerobic versus anaerobic POM consumption is an exciting avenue
for future work!

*Some particles, especially the ones that are a lot like spheres of chitin, might not have many microbes that*
*carry out cooperative processes on them, but many particles would likely still have some microbes*
*degrading them, even if none of them were exo-enzyme producing cooperators. In the presence of non-*
*cooperative degraders, my intuition is that you would rarely/never have local microbial extinctions on*
*particles, per figure 1A simulations A and B. However, I think the other processes, especially variability*
*in particle lability, would likely still have a big impact on flux.*

We completely agree that non-cooperative degradation occurs and that all particles have some microbial
degraders present on them. Our model simply predicts that many particles have struggling microbial
populations (defined as populations in sub-exponential growth) and that this is necessary to generate the
long tail of POM given realistic microbial dynamics. In Figure 1, we are comparing model and laboratory
results⁷ and so we do not include encounter dynamics. We also assume only extracellular enzyme
degradation to match the laboratory culture. In the real ocean, there will be encounters and so true
extinction would not occur in the same way that it occurred on the synthetic particle in the lab.

*I don't think this is a fatal flaw, since an inability of cooperative species to degrade particles could indeed*
*be a factor that decreases the remineralization of particles (and increases DOM flux), especially if*
*cooperative particle degradation ends up being the most efficient way to consume a particle. That said, I*
*think either more caveats, or if I have misunderstood the situation, counter arguments, would strengthen*
*the paper.*

Thank you very much for these constructive comments! We feel the additional model formulation and
new text (lines 155-160) have allowed us to strengthen our argument. We have also done an additional
calculation to show that extracellular enzyme degraders will experience sub-exponential (2nd order) rather
than exponential growth due solely to diffusive loss of LMWOM – in other words, we do not need to
invoke cooperation. By rearranging the growth, degradation, and diffusion equations we can derive the
following relationship where growth is equal to:

$$\mu = \mu_{max} \frac{\rho}{(\rho + \rho_c)}$$

where μ_{max} is the maximum growth rate, ρ is the bacterial density on the surface (cells m⁻²), and ρ_c gives
the critical cellular density below which bacterial growth is sub-exponential. ρ_c can be calculated as:

$$\rho_c = \frac{y_{lmwom} D k_m}{h \mu_{max} r}$$

where y is the yield, D is the diffusivity of the LMWOM, r is the radius of the particle, h is the width of
the diffusive boundary layer, and k is the half saturation constant for uptake of LMWOM. From this we
can estimate the time it takes to reach exponential growth (t_w) as:

$$t_w = \frac{1}{\beta y} \left[\frac{\rho_c - \rho_0}{\rho_0} - \ln(\rho_0 / \rho_c) \right]$$

where ρ_0 is the bacterial density at particle formation, and β is the polymer degradation rate. The time to
exponential growth can be plotted as a function of β and the ratio of ρ_0 / ρ_c .

 Note the axis values are a function of the choice of parameter values (here we selected values which
 matched a culture of *Vibrio splendidus* growing on chitin), but the shape of the plot is robust. Also, this
 calculation does not include attachment and detachment dynamics. This calculation highlights that the
 model simulations apply to any microbes that degrade POM through extracellular enzymes regardless of
 whether they are working cooperatively.

 *As a more technical consideration: I think figure 2A could perhaps be presented a little differently. I'm*
 *struggling to internalize the differences between the model runs A-G.*

*I get that the main point is that they have some runs where the microbes die out, some where they grow*
 *exponentially, and some where they struggle at first and then "recover". That said, I have trouble keeping*
 *track of which parameters are varying in which runs. I don't insist on this, but it would sure be nice if the*
 *authors could think about laying out the different parameter sets in a way where it is more intuitive for*
 *the reader to keep track of which things are different between which runs.*

Thank you for this suggestion. We have updated the Figure 2 caption to try to help with clarity and are
 open to any additional suggestions on how to further improve the legend and caption.

 *Also 2A: Why did they pick these parameter sets and not others? Is it because these give good examples*
 *of different microbial dynamics?*

Yes, we chose representative examples. We include all of the parameter combinations in *Supplement Fig.*
 *S23* but this is much harder to see as there are many lines.

 *One more question about 2A: panel 1: In the model runs C-G where the microbes do grow, their numbers*
 *appear to approach infinity. I presume there is some limit to microbial abundance on particles in this*
 *model. Why do these not appear to have any upper asymptote?*

In the current main text model simulations, we do not impose a carrying capacity and so the cells grow
 exponentially until the particle is fully consumed. Once the particle is consumed, they disperse into the
 water column. We have added an alternative version of Fig. 2a in the supplement (*Supplement Fig. S12*)
 which shows the full dynamics. The cellular density reaches 10^7 - 10^8 cells mm^{-2} which is consistent with
 observations^{7,13-15}.

 However, we agree that in reality there is an effective 'carrying capacity' for particles and that the
 abundance should asymptote. We have a version of the model where we imposed such a carrying capacity
 (we assume that the maximum cellular density is a mono-surface layer with no penetration). This model
 does result in asymptotic behavior as Dr. Cram anticipates. However, we found that such a formulation
 did not alter the model dynamics. This is because once the population is at the carrying capacity, the
 particle degradation rate is extremely rapid and so the particle is fully consumed very quickly. Sinking

speeds are also typically low because the particle is shrinking rapidly and so the impact on the POM flux
is negligible. As a result, we decided to use the simpler representation of microbial detachment dynamics
for the main text results. The results of the dynamic detachment model are discussed in *Supplement S2*.

*Figure 3: The gray lines tell us about the spread of all of the runs, but they don't tell as much about the*
*spread of the runs within the groups. I am wondering if it would be possible to add standard deviations,*
*or color code the individual runs so that we can see the within run spread of the different sets of runs.*

We have altered the colors of the lines to highlight the spread between the different sets of runs in Figure
3.

*3D: Is this relationship between encounter rate and flux at all dependent on the initial colonization density*
*of the particles? My hunch is that the encounter rate is likely less important if the microbes are pre-*
*colonized.*

Yes, you are correct that there is an important interaction between encounter rate, initial colonization
density, and flux. As you suspect, if the initial colonization density is above the critical threshold, the
particle is consumed rapidly (exponential growth). In this case neither how much above the threshold nor
the encounter rate matters much for the rate of particle degradation – our beta (particle lability) becomes
the most important factor. However, if the initial colonization density is below the critical threshold,
encounter dynamics can have a significant impact on particle degradation rate by rescuing the community
or at least sustaining it.

*Figure S4 -- Other publications have suggested a strong relationship between temperature and particle*
*flux (eg DeVries and Weber 2017) The relationship between temperature and flux seems to be minimal*
*when a Temp-Off (constant temperature) model is compared to a Temp-on one where temperature varies*
*through the water column. What if the temperature through the water column varies as seen in different*
*ocean regions? Maybe such analysis is beyond the scope of this paper. And indeed, it is technically true*
*as the authors say that "Temperature is not the primary driver of the observed dynamics" (lines 93-94),*
*since the observed dynamics only occur with the aforementioned temperature profiles. However, it's worth*
*noting that temperature is still likely an important driver of region-to-region particle flux, and that the*
*authors have not shown that temperature is unimportant, just that it doesn't matter much in this model*
*when only one temperature profile was considered.*

We have added additional text (lines 177 -191) to discuss this important point. Temperature is an important
physical constraint and plays a role in the 'loss' dynamics we describe in the paper. Specifically, because
temperature acts to reduce growth rates it becomes more challenging for communities to overcome the
other loss terms. However, we believe the temperature and particle flux relationship that has been
previously reported might in fact be driven in part by co-varying processes, namely shifts in lability and
microbial community composition. The default model includes a large temperature change over the water
column ranging from 26°C to 2°C (0-2,000 m) resulting in a 55% reduction in growth by 2,000 m.
However, if a population is able to grow exponentially, the impact of temperature becomes secondary. In
our model, temperature becomes important for the struggling communities which are associated with very
slow particle consumption rates and linear degradation. We have clarified this point in the text (lines 180-
187).

To more fully understand the role of temperature in the model, we have run several additional simulations.
We ran the model using a larger temperature-associated decrease in growth rate such that growth rate

decreased by 80% in the upper 2,000 m consistent with the formulation from Dr. Cram¹⁶, instead of 55%
in the default simulation. At 200 m, the differences in growth rates between the two model formulations
could be explained simply by the differences in temperature limitation (Fig. S33). By 500m, we see that
more populations have reached exponential growth in the milder-temperature limitation simulations
(overcome diffusive loss of LMWOM) than in the enhanced-temperature limitation simulations (reduced
growth due to lower LMWOM concentrations) (Fig. S33). However, despite these differences in growth
rates, we see that water column POM fluxes are very similar between the two runs (Fig. S19). We attribute
this to the fact that the model dynamics are not fundamentally changed: populations that are able to
successfully colonize particles and grow exponentially are able to still do so and the change in maximum
growth rate due to temperature limitation does not significantly alter the depth at which the particle is
completely consumed. When populations are struggling, their growth rates are low in both simulations,
the consumption of the particle is therefore also slow, and the particle persists.

To simulate a different oceanographic region, we have also run the model using a narrower temperature
gradient from 12°C to 3°C (0-2,000 m) based on a temperature profile from the subpolar North Atlantic
region (CCHDO Hydrographic Cruise 64PE20000926). We found the same degree of variability of POM
flux captured in both simulations (*Supplement S3, Supplement Fig. S18*).

*Line 119-120 “A struggling particle-associated population can be rescued by recruitment from the free-*
*living bacterial pool as the particle sinks...”*

*I see your examples of this happening (eg Figure 2 C and D), but I’m still not sure how this process works.*
*Do encounter rates sometimes increase with depth? I could imagine a population struggling and then the*
*particle sinks into a region with high encounter rates. And then the encounter rates exceed mortality until*
*the population reaches the threshold where cooperation between the microbes can break down the*
*particle. But if encounter rates scale with microbial abundances which are either most abundant at the*
*surface, or decreasing with depth, I’d assume the attachment would either exceed mortality at the surface,*
*in which case no “rescue” is necessary, or it wouldn’t and the population would crash. What am I missing*
*here?*

You are correct that encounter rates scale with microbial abundances in the water column and so encounter
rates are highest at the surface and decrease exponentially with depth (see figure below). The
compounding factor is that growth is limited by access to LMWOM, in the default model. In the surface,
when the initial population is below the critical threshold (i.e., struggling), the production of LMWOM is
low and growth is limited by the access to carbon. Encounter in the surface allows the population to persist
(overcome loss) but it takes some time before the population surpasses the critical threshold and can grow
exponentially. A secondary compounding factor is temperature. Both growth and mortality decrease with
temperature as the particles sink deeper in the water column, while encounter rate and detachment rates
are independent of temperature. This also impacts the timing (and depth) of the rescue of a struggling
population. In the new model simulations with direct uptake, ‘rescue’ populations are exceedingly rare as
either growth minus loss plus encounter is positive and the population reaches exponential growth, or it is
negative and the population struggles. In the rare instances where encounter rate is very close to growth-
loss, then the population will reach exponential growth but will be a very long lag phase due to very slow
net growth ($\frac{dB}{dt}$). As with the default model simulations, differential temperature limitation with depth also
impacts the dynamics in the direct uptake model.

The situation where encounter rates increase at a certain depth is exciting and one that we hope to explore
more in the future as this would add additional variability to POC degradation. However, in the current

model, we use a simple exponential decay of encounter rates with depth, resembling the fit of free living
microbial abundance profile with depth as measured by TARA ocean campaign¹⁷.

*Line 198-201: “We show that POM consumption is temporally and spatially variable as a function of*
*microbial growth dynamics... Supplement S1, S2, S3” -- I’m not sure I see how they show this. Are they*
*saying that dynamics are different from one particle to another and between depths? As written, this*
*implies that the authors have evidence that these factors vary between different ocean regions, or perhaps*
*seasonally, but I don’t think they tested such variability.*

We thank you for highlighting this point and have edited the text to clarify our meaning (lines 216-230).
Specifically, we know that microbial communities vary geographically and with depth and the type
(represented by lability and size) of particles vary regionally. We show that such changes have significant
and predictable impacts on the rate of POC degradation such that, in order to understand POC flux in the
ocean, we need to take into account these mechanisms.

*Line 211 “single type of bacterial primary degrader” -- One way that I misread this at first was that I*
*thought that the model assumed that there was one type of bacterial degrader *total*. But actually there*
*is one bacterial degrader *per particle* right? And those different degraders have different growth rates*
*and temperature sensitivities and so on. In either case, the authors might consider clarifying a bit more*
*in the text.*

Yes this is correct! One type per particle. We have edited the text to clarify this (lines 261-267).

*Reference:*
*DeVries T, Weber T. The export and fate of organic matter in the ocean: New constraints from combining*
*satellite and oceanographic tracer observations. Global Biogeochem Cycles. 2017 Jan 1;2016GB005551.*

*Signed,*
*Jacob Cram*

*Reviewer #2 (Remarks to the Author):*

*Review of Nguyen et al. for Nature Communications*

Thank you for this constructive review! We feel that addressing your comments have significantly helped
strengthen the paper and we would like to explicitly acknowledge your contribution in the
acknowledgements if you are OK with this.

*General*

*Given the lack of detailed mechanistic advances in teasing apart the different facets of the oceans'*
*biological pump this is a very timely study. It builds on conceptual papers from Aumont et al. (2017),*
*Zakem et al. (2021) and on some new Twilight Zone omics work from Ed de Long's lab (Foff et al., 2021).*
*It will be a useful stepping stone to improving our insights into how changing climate will influence the*
*oceans' biological pump – as stated in their Abstract.*

*There was one thread throughout the manuscript that I was surprised to see – one that was centred on*
*equating the “well-known POC flux profile” solely with microbial dynamics (rates, fluxes, community*
*structure). This profile is the cumulative imprint of microbes, zooplankton flux feeding, and at depth*
*(diurnal zooplankton migration) with little known about what the depth-dependency of each of these*
*processes are, their relative contribution to flux attenuation, and the implications of discrete (flux feeding)*
*versus (continuous) processes in setting this flux profile (see conceptual figure in Buesseler and Boyd,*
*2009, and also modelling studies by Jackson and Burd on flux feeding simulations).*

*Thus, there are issues around assuming that this flux profile is exclusively (or predominantly) set by*
*microbes and hence the value of looking for match-ups with this flux profile is questionable. I think the*
*authors can get around this impasse by exploring studies that report on the depth dependency of rates*
*(rather than community structure/ 16S ...) such as for enzymatic activities, remineralisation, etc. What*
*type of profiles do we see from such microbial processes.*

Thank you for highlighting this important point that needs clarification in our manuscript. We completely
agree that other processes, particularly those related to zooplankton, are also critical in setting the POC
flux. We have significantly revised the manuscript to specify that we are studying one aspect of POC
degradation and highlight that additional work is needed to couple our model on microbial processes with
explicit representation of zooplankton dynamics (lines 36-41, 44-53, lines 228-244). We now discuss how
our model framework can be used in future studies to answer some of the fundamental questions that Dr.
Boyd raises related to the relative importance of microbial versus zooplankton dynamics in setting the
POC flux profile (lines 228-244).

We feel that our model provides critical mechanistic insight into particle-associated microbial dynamics
and how this can translate into variability in POC degradation. Specifically, by moving away from ‘black-
box’ remineralization to more mechanistic representation of microbial ecology, we generate specific
hypotheses as to how microbial POC consumption should vary with depth and as a function of particle
composition. We are excited because we now have specific hypotheses that can be tested in the field to
help further improve the model and ultimately lead to mechanistic predictions of POC flux changes. Such
experimental work is needed because bulk measurements of carbon flux with depth provide a poor
constraint on the underlying mechanisms (as the reviewer highlights and we discuss on lines 234-258).

*Specific*

*Abstract*

*18 – crucial role – yes – but can we quantify it – given that many prior studies using imperfect tools*
*(leucine, thymidine etc) at atmospheric pressure do a poor job of balancing the C demand vs C availability*
*(inferred from the particle flux attenuation profiles).*

*19-22 – see general comments above.*

*23 – enhanced transfer where?*

*24 anthropomorphic terms – struggling !*

We have modified the text to provide a clear definition of ‘struggling community’. However, we would
like to keep this term as we feel it adequately describes the microbial communities which are “proceed[ing
to grow] with difficulty or with great effort” (Meriam-webster dictionary).

*25 what about bacterivory – see movies from Kiorboe lab and patterns of colonisation from this group.*

Thank you for highlighting this oversight. We now specify in the main text that bacterivory is one of the
important mortality loss terms for particle associated communities (lines 238-244) and that we need better
constraints on this term (lines 253-254).

*Introduction line 34 – what about zooplankton flux feeding – see Jackson and Burd modelling papers.*

We have modified the text to include this important process (line 36-38, lines 44-46, lines 231-244).

*39 – sinking speed and zooplankton mediated particle transformations.*

*40 – is it solely microbial POC consumption that drives these models – or tuning the ‘remineralisation’*
*as a bulk term?*

Yes you are correct, in the current generation biogeochemical models it is a bulk ‘remineralization term’
that drives POC consumption. We have edited the text to clarify this (lines 46-52).

*44 ‘recalcitrance is in the eye of the solubiliser’!! It’s a relative term – as we know from the soil C*
*literature.*

Yes! We agree ☺ we have added a few papers¹⁸⁻²¹ to support this statement.

*46 – please cross check these papers to see if they were done at ambient pressure (sea level). See Christian*
*Tamburini’s pressure work.*

We have added text (line 58 - 59) to highlight the potential impact of pressure on microbial activity and
particle degradation at depth. While we do not explicitly include the impact of pressure on microbial
growth in the model, we have now run several additional sensitivity studies with different temperature
limitation profiles including one with enhanced temperature limitation at depth. This enhanced limitation
with depth would produce similar results to the incorporation of limitation due to pressure (*Supplement*
*S3 and Fig. S19*). We demonstrate that the model findings are robust under these different limitation
scenarios.

*49 – what about the role of the diffusion boundary layer in these processes?*

Based on the seminal work of Kjørboe²²⁻²⁴ and others, we assume that LMWOM diffuses away from the
particle surface within the diffusive boundary layer and then is lost through advection at the diffusive
boundary layer boundary. We use the derivations of Kjørboe *et al* to include these microscale physical
dynamics into our model.

*59 – see general comments – this all needs careful re-consideration.*
*64/65 – ditto – needs much more stringent justification – see general comments*
*68 – ditto – revisit these after resolving the following from above.*
*“This profile is the cumulative imprint of microbes, zooplankton flux feeding, and at depth (diurnal*
*zooplankton migration) with little known about what the depth-dependency of each of these processes are,*
*their relative contribution to flux attenuation, and the implications of discrete (flux feeding) versus*
*(continuous) processes in setting this flux profile.”*
We have revised this text to address the general comments above.
*Results – their interpretation will need to be carefully revisited – see general comments above.*
We have revised this text to address the general comments above.
*84 – some sensitivity analysis on using a 100 m depth is essential – given that the stratum of de novo*
*particle synthesis can be much shallower (50 m or less and varies regionally) and that this region and the*
*underlying 50-100 m band of waters exhibit major gradients with decreasing stocks – then the depth where*
*microbial colonisation commence will be a key determinant of whether groups can attain critical particle*
*population sizes. This regional variation in this depth will influence the regional model comparisons again*
*using matchups with the POC flux profiles – which needs to be reconsidered.*
Thank you for highlighting this important point that we did not previously address sufficiently. We chose
100 m as this is classically used as the particle export depth²⁵. However, we agree that this is not fixed
and will vary. We have conducted an additional set of simulations where we release the particles at 50 m
and 75 m. This impacts encounter dynamics and the temperature function. These new results nicely
demonstrate how variable particle formation depths can significantly impact the POM flux in the surface
ocean. However, we see convergence in the model results below ~500 m. We hypothesize that regional
differences in POC flux profiles between 500 and 2,000 m are primarily driven by differences in particle
type (lability), particle size spectra, and microbial communities while differences above 500 m will be
strongly impacted by formation depth. We have added text describing these additional simulations (lines
177-191) and include a more complete description and figures in the supplement (*Supplement S2*, and
*Supplement Fig. S20*).
*106 – an intriguing finding – worthy of a mention in the Abstract?*
*109 – half of the time?? Vague*
We have clarified this phrase.
*110 deep ocean?? Vague*
We have clarified this phrase.
*112 – see comment above on arbitrary 100 m initialisation depth.*
We have added new text about variable particle export depths (lines 101-103, lines 189-191, lines 272-
274).
*118 struggling?*
*120 rescued?*
*121-124 – see general comments above.*
*127 thriving struggling??*
We have added text clarified our terminology.

give an example of a single particle type please

We have clarified this phrase.

– se comment on the need for regional variations in the (100 m) initialisation depth.

We have added new text about variable particle export depths (lines 189-191, lines 272-274).

to 144 – this demarcation is a ‘self-fulfilling’ outcome ?

We agree that this outcome makes sense (is an intuitive result of the model). But we also felt it is a nice illustration as to how our model can make predictions about microbial dynamics which can be directly tested in the field.

– some of these references refer to model communities

We have revised this list of references so that they are all for experimental studies.

– in particular – see general comments above!!

We have revised this text to address the general comments above.

Discussion – will need to be overhauled in line with General comments above.

For example line 184

We have revised this text to address the general comments above.

I identify myself as a reviewer – please pass on my name to the authors

Thanks

Philip Boyd

**Reviewer #3 (Remarks to the Author):**

*The vertical flux of POC in the ocean decreases roughly exponentially with depth. This depends on a great*
*many factors, including, for example, surface ocean production, community composition, particle*
*characteristics, mineral ballast, the mineralization of organic matter, and everything that impacts those.*
*Nguyen and colleagues contribute to the discussion of the processes that controls this flux by presenting*
*a numerical model that focuses on the dynamics of heterotrophic bacteria that is responsible for POM*
*mineralization. A strong motivation for their work is the desire to connect the observed POC fluxes with*
*the underlying mechanisms acting at the particle level, thereby improving the ability to capture and*
*explain the observed variability and ultimately the predictions of the future biological pump. The*
*manuscript is well written, timely and important.*

*The authors demonstrate that POM mineralization rates can be high and variable due to the underlying*
*microbial dynamics, producing results consistent with the observed flux - depth relationships. In their*
*model, POM degradation progresses via the formation of low molecular weight (LMW) OM, which then*
*is either consumed or lost from the particle to the surrounding water and is assumed to be at steady state.*
*Bacterial biomass is affected by growth, with growth rates depending on growth efficiencies, and*
*LMWOM concentrations, death, detachment and encounter rates. The model emphasis on microbial*
*dynamics follows in the footsteps of Mislan et al. 2014, but it expands on it by looking at a number of*
*aspects relevant for the biological pump, for example particle size distributions etc. Here, the particle*
*radius (selected from realistic particle size distributions) sets the initial POM, which is the modified due*
*to reaction, reducing particle diameters and sinking speed.*

*There are many ways to approximate each of the processes considered, but I found the formulations*
*chosen to be reasonable and well informed by existing work. Importantly, the authors provide a detailed*
*assessment of the impact of some of the key assumptions made when translating the complex system of*
*sinking particles and associated microbial community into a very simple set of equations. The model is*
*described clearly and relatively easy to follow, and it nicely includes the consideration of fractal*
*dimensions and hydrodynamics in the parameterization. In several instances, when different*
*parameterizations seemed reasonable or a conceptual approach gave me pause, the authors demonstrated*
*through sensitivity tests that the impact on the outcome is small (supplement S2; examples include the*
*particle size distribution, the one microbial community per particles etc.).*

**Thank you very much for this constructive review!**

*Some suggestions for potential improvements:*

*- organic mineralization rate: POM to LMWOM transformation is proportional to a cell specific rate beta*
*and the cell density of the heterotrophic bacteria on each particle. There is no explicit dependency on the*
*amount of organic matter (e.g. Gloege et al. 2017 Global Biogeochem. Cycles 31, 1192-1215; Aumont et*
*al. 2017, Biogeosciences 14, 2321-2341). This reflects a particular conceptual model on the rate limiting*
*steps I consider reasonable, and the impact of OM is simulated via a feedback on the bacterial cell*
*numbers. Furthermore, the tests considering single vs. 25 lability types in a particle (S2) seem to suggest*
*that the timescales are such that a potential POM dependency is captured through this mechanism. But*
*have you considered POM-dependent kinetics and if so, can you confirm that the outcome is not sensitive*
*to this selection? And in the model, do you continue to consume POM when POM is 0 but $B > 0$ (eq. 1; is*

*B considered part of POM?) What kind of bacteria/POM ratios are you obtaining in the simulations?*
*(Line 242: point to eq. 11, not 13; in eq. 5 consider a discrete summation)*

Thank you for this suggestion. The original model only represents one type of microbial degradation –
extracellular cleavage of polymers into LMWOM. While we believe this is the dominant form of microbial
polymer degradation, other degradation processes are possible including direct uptake of the polymer (i.e.
selfish-uptake⁸⁻¹⁰). To understand the impact that this alternative uptake strategy might have on particle
degradation and the findings of our paper, we have developed a modified version of the model that
simulates direct uptake by the degraders (i.e., no LMWOM is generated) (described in *Supplement S3*).
We ran this model using a similar range of parameter values (initial biomass, free living concentration,
particle size, etc.). Lability in this model must be treated differently – here we modify the beta term in the
original model to ξ which is constrained by the rate at which microbes can incorporate carbon into their
biomass (i.e., microbes cannot degrade the particle faster than they can grow). We believe this is what the
reviewer was referring to with POM-dependent kinetics. We demonstrate that the three types of particle
degradation profiles also emerge for direct-uptake degraders: 1) exponential growth and rapid
consumption, 2) populations struggling to overcome loss (here only mortality and detachment rate), and
3) rescued populations due to encounter with free-living bacterial communities (as shown in *Supplement*
*Fig. S14*). Of note, the ‘rescued’ populations are more common in the enzymatic cleavage model than in
the direct uptake model.

This alternative mechanism for POM degradation does not change the primary finding of our works: POM
consumption is temporally and spatially variable as a function of microbial growth dynamics and
microbial community composition. Changes in microbial processes (e.g., enzyme kinetics, loss rates) have
the potential to significantly alter the rate of particle degradation and thus the POM flux. We describe this
new model in *Supplement S3*. We have added additional text (lines 155-160) and supplement figures
(*Supplement Figs. S13-15*) summarizing these new model findings. We also have added text highlighting
the need to constrain loss processes for these communities as this is the key driver of whether the
populations successfully colonize particles or struggle to survive (lines 252-254).

We also wish to highlight that both model formulations use a fractal representation of particle such that
particle density and carbon content per volume vary with particle size (please see *Supplement S1* for a full
description of this formulation).

Thank you for pointing out that we neglected to specify how we deal with complete particle consumption.
When the particle is fully consumed (POM=0), the particle-attached bacterial community detach and so
consumption stops. We have added this important detail to the methods (lines 283-285). We also ran
model simulations with a more complicated detachment formulation where bacterial detachment rate
increases as the particle shrinks so as to keep the bacterial density on the particle close to a mono-layer of
bacteria. However, we demonstrate that the model dynamics are the same for both formulations and so
opt to use the simpler formulation of a constant leave rate.

The model predicts that a range of ratios from 0.002 cells per particle to 10^8 cells per particle which is
consistent with the data that we were able to find^{7,13-15}.

We apologize for the typo in the equation references, we have corrected this and also corrected eq. 5 to be
a discrete summation.

- missing processes: Particle aggregation and disaggregation or zooplankton grazing are generally considered to be important and have improved model fits to fluxes, yet are not considered here (e.g. Karakas et al. 2009, Progress in Oceanography 83, 331-341; Burd and Jackson 2009 Annu. Rev. Mar. Sci 1, 65-90). Zooplankton dynamics can contribute to the spatial and temporal variability in fluxes (e.g. Jackson and Burd 2002, DSR II, 49, 193-217; Cavan et al. 2017 Nat. Comm 8:14847, considering the effect of zooplankton in the context of OMZs; Boeuf et al. 2019. PNAS 116, 11824-32, suggesting that “a considerable amount of sinking POM degradation may be due to its consumption by mid- and deep-water animals, and their associated deep-sea adapted microbiota”), and that the formation and breaking up of aggregates are critical for the determination of sinking velocities, time for remineralization and hence C fluxes.

The nagging question is to what extent ignoring these processes yet matching the Martin curve overemphasizes the role of the processes highlighted here. To some extent, the authors address this in the Supplement S3, but a clear(er) discussion of what measurements are needed to assess the relative importance of microbial vs. the aggregation/disaggregation processes on fluxes would be helpful. Consider to explicitly identify the specific measurements your model results indicate that will distinguish the numerous processes that may explain observed fluxes (fig 3), high growth rates and variable POM consumption rates (line 66ff). This would further bolster the comments made regarding the need for better in situ measurements (paragraph line 190ff) and testable hypotheses (line 69).

Thank you for this suggestion! We have modified the text to include a more complete description of zooplankton dynamics and the impact on POM flux (lines 36-41, 44-53, lines 228-244). We also now discuss how our model could help tease apart the different drivers of the POM flux and generate hypotheses which can be tested in the field (lines 228-244).

**References**

- 1. Arnosti, C. Patterns of Microbially Driven Carbon Cycling in the Ocean: Links between Extracellular
Enzymes and Microbial Communities. *Adv. Oceanogr.* **2014**, e706082 (2014).
- 2. Berlemont, R. & Martiny, A. C. Glycoside Hydrolases across Environmental Microbial Communities.
*PLOS Comput. Biol.* **12**, e1005300 (2016).
- 3. Kabisch, A. *et al.* Functional characterization of polysaccharide utilization loci in the marine
Bacteroidetes ‘Gramella forsetii’ KT0803. *ISME J.* **8**, 1492–1502 (2014).
- 4. Weiss, M. S. *et al.* Molecular architecture and electrostatic properties of a bacterial porin. *Science*
**254**, 1627–1630 (1991).
- 5. Nguyen, T. T. H., Myrold, D. D. & Mueller, R. S. Distributions of Extracellular Peptidases Across
Prokaryotic Genomes Reflect Phylogeny and Habitat. *Front. Microbiol.* **10**, (2019).
- 6. Arnosti, C. Microbial extracellular enzymes and the marine carbon cycle. *Annu. Rev. Mar. Sci.* **3**,
401–425 (2011).
- 7. Ebrahimi, A., Schwartzman, J. & Cordero, O. X. Cooperation and spatial self-organization determine
rate and efficiency of particulate organic matter degradation in marine bacteria. *Proc. Natl. Acad. Sci.*
(2019) doi:10.1073/pnas.1908512116.
- 8. Reintjes, G., Arnosti, C., Fuchs, B. & Amann, R. Selfish, sharing and scavenging bacteria in the
Atlantic Ocean: a biogeographical study of bacterial substrate utilisation. *ISME J.* **13**, 1119–1132
(2019).
- 9. Reintjes, G., Fuchs, B. M., Amann, R. & Arnosti, C. Extensive Microbial Processing of
Polysaccharides in the South Pacific Gyre via Selfish Uptake and Extracellular Hydrolysis. *Front.*
*Microbiol.* **0**, (2020).

- 10. Arnosti, C., Reintjes, G. & Amann, R. A mechanistic microbial underpinning for the size-
reactivity continuum of dissolved organic carbon degradation. *Mar. Chem.* **206**, 93–99 (2018).
- 11. Ploug, H. Small-scale oxygen fluxes and remineralization in sinking aggregates. *Limnol.*
*Oceanogr.* **46**, 1624–1631 (2001).
- 12. Bendtsen, J., Hilligsøe, K. M., Hansen, J. L. S. & Richardson, K. Analysis of remineralisation,
lability, temperature sensitivity and structural composition of organic matter from the upper ocean.
*Prog. Oceanogr.* **130**, 125–145 (2015).
- 13. Rieck, A., Herlemann, D. P. R., Jürgens, K. & Grossart, H.-P. Particle-Associated Differ from
Free-Living Bacteria in Surface Waters of the Baltic Sea. *Front. Microbiol.* **6**, (2015).
- 14. Lampitt, R. S., Wishner, K. F., Turley, C. M. & Angel, M. V. Marine snow studies in the
Northeast Atlantic Ocean: distribution, composition and role as a food source for migrating plankton.
*Mar. Biol.* **116**, 689–702 (1993).
- 15. Möller, K. O. *et al.* Marine snow, zooplankton and thin layers: indications of a trophic link from
small-scale sampling with the Video Plankton Recorder. *Mar. Ecol. Prog. Ser.* **468**, 57–69 (2012).
- 16. Cram, J. A. *et al.* The Role of Particle Size, Ballast, Temperature, and Oxygen in the Sinking
Flux to the Deep Sea. *Glob. Biogeochem. Cycles* **32**, 858–876 (2018).
- 17. Sunagawa, S. *et al.* Tara Oceans: towards global ocean ecosystems biology. *Nat. Rev. Microbiol.*
**18**, 428–445 (2020).
- 18. Schmidt, M. W. I. *et al.* Persistence of soil organic matter as an ecosystem property. *Nature* **478**,
49–56 (2011).
- 19. Dittmar, T. *et al.* Enigmatic persistence of dissolved organic matter in the ocean. *Nat. Rev. Earth*
*Environ.* **2**, 570–583 (2021).

- 20. Lehmann, J. *et al.* Persistence of soil organic carbon caused by functional complexity. *Nat.*
*Geosci.* **13**, 529–534 (2020).
- 21. Zakem, E. J., Cael, B. B. & Levine, N. M. A unified theory for organic matter accumulation.
*Proc. Natl. Acad. Sci.* (2021) doi:10.1101/2020.09.25.314021.
- 22. Kiørboe, T., Ploug, H. & Thygesen, U. H. Fluid motion and solute distribution around sinking
aggregates I : Small-scale fluxes and heterogeneity of nutrients in the pelagic environment. *Mar.*
*Ecol. - Prog. Ser.* **211**, 1–13 (2001).
- 23. Kiørboe, T. & Jackson, G. A. Marine snow, organic solute plumes, and optimal chemosensory
behavior of bacteria. *Limnol. Oceanogr.* **46**, 1309–1318 (2001).
- 24. Kiørboe, T., Grossart, H.-P., Ploug, H. & Tang, K. Mechanisms and Rates of Bacterial
Colonization of Sinking Aggregates. *Appl. Environ. Microbiol.* **68**, 3996–4006 (2002).
- 25. Cael, B. B. & Bisson, K. Particle Flux Parameterizations: Quantitative and Mechanistic
Similarities and Differences. *Front. Mar. Sci.* **5**, (2018).

The effect of microbial activities on ocean-scale carbon transport

Trang T. H. Nguyen¹, Emily J. Zakem¹, Ali Ebrahimi², Julia Schwartzman², Tolga Caglar³, Uria
Alcolombri⁴, François J. Peaudecerf⁴, Terence Hwa³, Roman Stocker⁴, Otto X. Cordero², Naomi
4 M. Levine^{1,(*)}

¹ Department of Biological Sciences, University of Southern California, Los Angeles, CA 91011,
USA

² Ralph M. Parsons Laboratory for Environmental Science and Engineering, Department of Civil
and Environmental Engineering, Massachusetts Institute of Technology, Cambridge, MA 02139,
USA

³Department of Physics, University of California at San Diego, La Jolla, CA 92093, USA

⁴ Institute of Environmental Engineering, Department of Civil, Environmental and Geomatic
Engineering, ETH Zurich, 8093 Zurich, Switzerland

(*) Corresponding author: n.levine@usc.edu

Abstract

**Sinking particulate organic matter (POM) out of the surface ocean sequesters carbon on**
**decadal to millennial timescales. Predicting POM fluxes is therefore critical for**
**understanding both global carbon cycling and the climate. Microbes play a crucial role in**
**POM degradation, but their associated impacts on ocean-scale POM fluxes are poorly**
**understood. Here we scale-up essential features of microbial dynamics to understand large-**
**scale vertical carbon transport in the ocean using a novel ecological model. The model**
**provides mechanistic insight into the well-known POM flux profile and observed shifts in**
**microbial communities with depth. We show that the enhanced transfer of carbon to depth**
**can result from populations struggling to establish colonies on sinking particles due to**
**diffusive nutrient loss, cell detachment, and mortality. These dynamics are controlled by the**
**interaction between multiple biotic and abiotic factors. Accurately capturing particle-**
**microbe interactions is essential for predicting variability and changes in large-scale carbon**
**cycling.**

Introduction

The vertical flux of carbon in the ocean has been the subject of numerous field campaigns,
laboratory experiments, and modeling studies over the past four decades^{e.g., 1–25}. Ultimately, the
vertical flux of POM is determined by the rate of POM production, the sinking speed of particles,
and the rate of POM consumption. Previous carbon cycle modeling studies aimed at understanding
large-scale fluxes have focused primarily on physical and chemical processes^{5,7,8,26} and have
shown that differences in particle size spectra^{5,9,16,17}, lability⁶, and temperature^{8,27} play important
roles in determining the efficiency of POM transfer, consistent with observations^{4,28,29}. However,
these large-scale models have not explicitly included microbial ecological dynamics. This is
problematic as many of the aforementioned physical and chemical mechanisms shown to impact
the vertical flux of carbon are controlled by microbial and grazer dynamics^{22–25,30}. For example,
lability as represented by these models is inherently related to microbial activity (as we

demonstrate below). Even the size of particles and thus their sinking speed is strongly influenced by the rate of degradation both by microbes and zooplankton dynamics.

Organisms alter the POM flux through a variety of processes. Heterotrophic microbes directly consume organic matter within particles with an estimated contribute of 70-92% of POM remineralization²⁵. Grazers such as zooplankton contribute to the aggregation, disaggregation, and consumption of particles^{23,25,30-37} with small particles acting as a primary food source in the mesopelagic ocean³³. However, rather than explicitly capturing these dynamics, current biogeochemical models rely on low and invariant rates of POM consumption (typically using bulk remineralization terms) tuned in order to match observed POM flux profiles^{5,6,8}, rather than mechanistically representing variable rates of POM degradation driven by microbial and grazer dynamics. Here we focus on the mechanisms behind the persistence of particulate organic matter in the deep ocean and the role that heterotrophic microbes play in altering the vertical flux of particulate carbon into the deep ocean. Additional work is needed to understand the coupled impact of microbial and grazer dynamics on the POM flux (see Discussion below).

Particle-associated heterotrophic microbial communities are incredibly dynamic³⁸⁻⁴¹ and able to achieve relatively rapid growth rates (order of 1 to 10 day⁻¹)^{39,40,42-44} even on seemingly ‘recalcitrant’ organic carbon compounds^{38,41}. As a result, particles can be fully consumed on the timescales of days^{38-41,44}. This contrasts with observations of ‘labile’ particles in the bathypelagic and epibenthic^{21,45}. Previous work suggests that temperature and pressure limitation on microbial activity might allow for the persistence of particles in the deep ocean^{16,46-48}. In addition, microbial dynamics such as enzyme production, attachment, detachment, and mortality have been shown to play a key role in the rate of POM degradation^{38-40,43,49-52}. Evidence suggests that, for successful colonization of a particle to occur, particle-associated microbes must surpass the critical population size necessary for countering the diffusive losses of extracellular enzymes that are used to break down polymers (the main component of POM) and the resulting low-molecular-weight degradation products^{39,43,53-55}. These micro-scale observations suggest that the rate of POM consumption in the ocean is highly variable and a function of microbial processes, in contrast to the conventional representation in carbon cycle models^{42,43,51,56,57}. Therefore, to mechanistically understand the vertical flux of carbon in the ocean and generate robust predictions of future changes, we must account for particle-associated microbial behavior^{22,24,58}.

We present a water-column model that for the first time explicitly accounts for micro-scale observations, reconciles rapid microbial growth rates with slow POM remineralization timescales (order of 10⁻³ to 10⁻¹ day⁻¹)^{3,4,29,59}, and determines the impact of shifts in microbial dynamics on the rate of POM flux attenuation. Specifically, we identify key aspects of particle-associated microbial community dynamics that can generate the typical attenuation profile of observed POM flux and predict how changes in microbial community dynamics can rapidly shift rates of POM remineralization. This work demonstrates that the assumption of low constant POM consumption rates is incorrect, and that microbial dynamics alone can generate significant variability in the POM flux. Our results challenge the classic idea of particles being inherently labile or recalcitrant and propose that this is an emergent ecosystem property.

One of the difficulties in studying the vertical carbon flux in the ocean is that POM flux observations provide a poor constraint on models¹³ – in fact only 2 free parameters are needed to represent the classic POM flux curve while mechanistic carbon-cycle models rely on many more parameters. Thus, capturing observed POM flux profiles is not in of itself sufficient validation of

84 a proposed mechanism. Our model allows for targeted hypotheses that can be tested in the field
and will provide enhanced constraints on carbon-cycle models allowing for more robust
predictions of future changes in carbon export.

**Results**

Our mechanistic model captures the colonization of particles by microbes and conversion from
polymeric to low-molecular-weight organic matter (LMWOM) compounds as the particles sink
(*Methods* and *Supplement S1*). To represent the complexity of the interactions between diverse
microbial communities and a chemically diverse organic carbon pool, we use lability as an
ecosystem property that specifies the conversion rate of polymeric material to LMWOM of a
specific POM pool by a specific microbial group⁶⁰. As lability is poorly constrained by
observations, we test a wide range of POM lability values assuming a log-normal distribution⁶⁰,
though the modeled POM fluxes at the ocean-scale is found to be independent of the specific form
of the distribution chosen (*Supplement Fig. S1*). We represent the organic matter content of each
model particle using a single, stochastically assigned, lability value and represent the particle-
associated microbial community using a single microbial group per particle (in all we resolve 18
microbial types). Sensitivity tests with complex particles and multiple microbial groups per
particle and greater numbers of microbial groups show that these simplifications do not impact the
overall POM flux (*Supplement Figs. S2 and S3*). The model is initialized using a particle export
depth between 50 and 100 m (with a default value 100 m) with particle size distributions spanning
the observed range (power law with exponent $s = -2, -3, \text{ or } -4$; *Supplement Table S1*). Each particle
is stochastically assigned an initial radius and set of biological parameter values using a uniform
distribution (maximal growth rate, initial biomass, and fraction of the free-living microbial pool
that is capable of degrading the particle; *Methods*). The sinking speed of each particle is calculated
based on its size and density^{5,16,28,61}. As each particle sinks through the water column, microbial
growth on the particle evolves prognostically, and organic carbon is consumed or lost to the
surrounding water column due to diffusion and advection⁵⁴. The microbially-catalyzed
degradation of the particle causes the radius to shrink and the sinking speed to decrease.
Temperature dependence of microbial growth rates is applied using a typical water column
temperature profile⁶² (*Supplement Fig. S4*).

The model captures the observed density-dependent growth of particle-associated microbial
populations, where the population density must surpass a threshold in order for successful particle
colonization to occur⁴³ (Fig. 1b). Specifically, microbial populations must overcome multiple
sources of loss including mortality, detachment, and the diffusive and advective loss of LMWOM
away from the particle surface (hereafter referred to as loss processes). Model populations below
a critical threshold cannot establish a colony on the particle due to these loss processes, consistent
with laboratory observations⁴³ (Fig. 1b and *Supplement Fig. S5*). This population-size dependence
(the ‘Allee effect’⁵³) arises in our model due to the interplay between diffusive nutrient loss,
saturating (Monod) growth kinetics, and mortality (*Supplement Figs. S6-S9*).

[revised manuscript text omitted]

the mean, range, and distribution of observed POM flux profiles from across the global oceans
(n=897)³ (Fig. 3) suggesting that variability in microbial dynamics can play a significant role in
variable POM fluxes. Higher b values are associated with higher abundance of small particles
(flatter particle size spectra) and higher lability. Thus, oligotrophic regions with flatter particle size
spectra are predicted to coincide with higher b values, consistent with previous studies^{5-9,16}. As
expected, differences in particle size plays a key role in the efficiency of carbon transfer as larger
particles sink substantially faster (Fig. 2 and *Supplement Fig. S20*). However, we show that the
particle-associated microbial community dynamics can alter the POM flux to the same extent as a
change in particle size (Fig. 2d, *Supplement Fig. S22*). These model dynamics (from rapidly
growing populations to those struggling to survive) emerge as a result of stochastic interactions
between biological, chemical, and physical controls on microbial growth.

**Discussion**

[revised manuscript text omitted]

**Methods**

This model captures key micro-scale dynamics occurring on particulate organic matter (POM) in
 a manner scalable to the water column. For the results presented in the main text, we represent
 microbial diversity using 18 groups of particle-associated microbes, defined based on their enzyme
 kinetics and growth rates. Supplement S1 presents a sensitivity test with a continuum of microbial
 classes and demonstrates that this discretization does not impact our results. We also make the
 simplifying assumption that each particle type consists of a single lability of organic matter
 colonized by a single type of microbial primary degrader, though the conclusions of this work are
 not dependent on this assumption (*Supplement Fig. S2*). Here we track each unique particle type i ,
 which is defined based on radius at formation and lability (β_i), separately. We include enzymatic
 degradation of polymer into low molecular weight organic matter (C_{lmwom}), density dependent
 growth of heterotrophic microbes (B_i), and attachment ($E_{i,z}$) and detachment (L_i) of heterotrophic
 microbes to the particles. This model can be coupled to a full ecosystem model such that the
 generation of each particle type can be calculated prognostically. However, here we focus on the
 degradation of POM below the export depth and so simply include a source term to represent
 particle formation at 100 m (*see Supplement S2 for simulations with alternative formation depths*).
 An extended model description is provided in Supplement S3.

 The change in the carbon content of particle i ($C_{pom,i}$, mmol C_{pom} particle⁻¹) over time is defined
 as:

$$279 \quad \frac{dC_{pom,i}}{dt} = -\beta_i B_i \quad \text{eq. 1}$$

where β_i (mmol C_{pom} mmol C_{cell} ⁻¹ day⁻¹) represents the polymer degradation rate of $C_{pom,i}$ by
 microbial population B_i (mmol C_{cell} particle⁻¹) similar to⁵⁸. Specifically, β_i captures differences
 in ‘lability’ of particles, which is a function the organic matter itself, the microbial enzymes
 specific to group B_i , and production rate of those enzymes by B_i . When the particle is fully
 consumed (\$C_{pom}=0\$ ), the particle-associated microbial community detach and so consumption
 stops.

 The enzymatic degradation of POM results in the production of low molecular weight organic
 matter (LMWOM) (C_{lmwom} , mmol C_{lmwom} m⁻³ particle⁻¹) which supports microbial growth. We
 assume that there is no loss of carbon during the enzymatic cleavage from POM to LMWOM such
 that 1 mmol \$C_{pom}\$ degraded = 1 mmol \$C_{lmwom}\$ produced – there is however diffusive loss of
 LMWOM away from the particle as described in eq. 2. LMWOM concentration is calculated as:

$$292 \quad C_{lmwom} = (\beta_i B_i - \frac{\mu_i B_i}{y_{lmwom}}) / d_{loss} \quad \text{eq. 2}$$

where steady state dynamics are assumed, μ_i is the growth rate of microbes B_i on the particle (day⁻¹)
 y_{lmwom} is aerobic microbial growth efficiency (mmol C_{cell} mmol C_{lmwom} ⁻¹), and d_{loss} is the
 diffusion loss rate of LMWOM (m³ day⁻¹) (*Supplement S1 eq. s3*).

 Microbial dynamics on each particle are defined as:

$$298 \quad \frac{dB_i}{dt} = B_i(\mu_i - L_i - m_{lin,i}) + E_{i,z} \quad \text{eq. 3}$$

where $m_{lin,i}$ is microbial mortality rate (day⁻¹). L_i is the detachment rate (day⁻¹). The microbial
 encounter rate ($E_{i,z}$, mmol C_{cell} day⁻¹ particle⁻¹) varies with depth based on particle size, sinking
 speed, and the abundance of free-living and motile microbes B_i in the water column^{49,80} (eq. s11

in *Supplement S1*). Microbial growth rate, μ_i (day^{-1}) is dependent on the LMWOM concentration
 at the particle surface and is represented with the Monod equation:

$$304 \quad \mu_i = V_{max,i} \frac{C_{lmwom}}{C_{lmwom} + k_{m,i}} \gamma_{T,z} \quad \text{eq. 4}$$

where $V_{max,i}$, $k_{m,i}$, and $\gamma_{T,z}$ represent maximum LMWOM uptake rate, half saturation of LMWOM
 uptake of microbial group $B_{i,z}$ and temperature limitation at depth z (eq. s11 in *Supplement S1*),
 respectively. Model parameter values are given in *Supplement Table S1* and sensitivity tests are
 described in *Supplement S2*.

 The total POM flux at a certain depth z , $F_{pom,z}$ ($\text{mmol } C_{pom} \text{ m}^{-2} \text{ day}^{-1}$), is calculated as the
 accumulation of the vertical fluxes of each individual particle as they sink through the water
 column, where $F_{pom,z}$ is:

$$313 \quad F_{pom,z} = \sum C_{pom,i,z} N_{i,z} \omega_{i,z} \quad \text{eq. 5}$$

where $N_{i,z}$ is the number of particle type i per m^3 water column at depth z .

 To test the impact of particle associated microbial dynamics on the POM vertical flux through the
 water column, we perform a set of stochastic simulations in which parameters are randomly chosen
 from within a reasonable range. Here we simulate a single $2.24 \text{ m} \times 2.24 \text{ m}$ column initialized
 between 50 and 100 m with an observed particle size distribution yielding a total flux of 1.5 mol
 $\text{m}^{-2} \text{ yr}^{-1}$ ^{1,5,13}. Specifically, we simulate 2,256 particles (particle size spectra with power law
 exponent $s = -2$), 69,000 particles ($s = -3$), or 763,615 particles ($s = -4$) ranging from $50 \mu\text{m}$ to 0.4
 322 cm in radius as they fall through the water column. For each particle, the following model
 parameters are stochastically assigned within a reasonable range (*Supplement Table S1*) using a
 uniform distribution, except for lability for which a log-normal distribution is used: maximum
 growth rate V_{max} (1.2 or 7.2 day^{-1}), particle lability β ($10 - 1,000 \text{ mmol } C_{pom} \text{ mmol } C_{cell}^{-1} \text{ day}^{-1}$),
 initial cell density ($400 - 2,800 \text{ cell mm}^{-2}$), and density of free-living community of microbes B_i
 (F_i) (10 to 285 cell mm^{-3}). The stochastic simulations are conducted 1,000 times for each particle
 size distribution. Simulations are run for 600 days which is sufficient for all particles to be fully
 consumed or exported to $> 4,000 \text{ m}$. The attenuation exponent b for the modeled POM flux is
 calculated using a least square fit of the power law function.

**Acknowledgement**

This work was supported by a grant from the Simons Foundation (542387 to NML, 542395 to RS
and OC, 542389 to TW). We thank N. Norris and E. Lee for assistance with model development.
We thank our reviewers (Dr. J. Cram, Dr P. Boyd, and an anonymous reviewer) for their
constructive feedback.

**Author contributions**

338 T.T.H.N and N.M.L designed the study, developed the model, and conducted the analysis; E.J.Z,
O.X.C, A.E, J.S, U.A., F.J.P, T.H, and R.S contributed data and to model development; all authors
contributed to the writing.

**Competing interests**

The authors declare no competing interests.

**Data availability**

No new data is presented in this paper.

**Code availability**

[revised manuscript text omitted]

- 532

REVIEWER COMMENTS

Reviewer #1 (Remarks to the Author):

I thank the authors for their very thorough response to my review. I am impressed with their additional analysis and improvements to their figures and text. I am satisfied with their response and encourage the editors to accept this manuscript. This continues to be a great paper, made even better by the authors' additional work, and I am eager to see it published. The authors are welcome but not required to acknowledge my feedback, as they suggest.

I have one minor suggestion regarding Figure 3, which the authors may take or leave. I was initially confused when I first looked at the revised version of this figure with the different shades of grey lines indicating different runs with the same parameters because the averages of some of the lines appear to be at the edge, rather than close to the middle of the range of runs (Figure R1 -- Uploaded as PDF with this review).

I think what is happening here is that the dark gray lines are plotted first, then the medium grey, then the light grey. This means that the dark lines are concealed by the medium grey lines. This leads the reader to believe the range of values is narrower than the mean value. Such a problem could possibly be solved (I think) if the plotting function randomized the order in which it plotted the different line colors. Alternatively, my suggestion might instead make everything even harder to read. I think readers can figure out what is actually going on as written, so it's fine to leave it if my suggestion doesn't help.

All the best,
Jacob

Reviewer #2 (Remarks to the Author):

See 2 attached documents
Philip Boyd

Reviewer #3 (Remarks to the Author):

I appreciate the detailed response to the review comments, including the additional simulations. Some of the potential shortcomings or unknown effects (e.g., temperature, different paths of OM degradation) have been addressed. Other missing pieces are clearly identified, most importantly the lack of accounting for zooplankton; particle aggregation/disaggregation. The latter remains a significant question mark, albeit one that goes well beyond the scope of this paper. I consider this manuscript a very interesting addition to the discussion on the controls of POM fluxes, as it connects processes at the

particle scale with POM fluxes and highlights the role of microbial dynamics.

typos, minor comments

Line 43 - should be "contribution"

Caption figure 3: define UVP

Supplement 3, line 337: consider expanding, and point to the relevant equations you use to derive eq. s22-s24

Supplement 3, line 442 - s23 instead of s26

REVIEWER #1 (Remarks to the Author):

I thank the authors for their very thorough response to my review. I am impressed with their additional analysis and improvements to their figures and text. I am satisfied with their response and encourage the editors to accept this manuscript. This continues to be a great paper, made even better by the authors' additional work, and I am eager to see it published. The authors are welcome but not required to acknowledge my feedback, as they suggest.

Thank you again for your incredibly helpful comments!

I have one minor suggestion regarding Figure 3, which the authors may take or leave. I was initially confused when I first looked at the revised version of this figure with the different shades of grey lines indicating different runs with the same parameters because the averages of some of the lines appear to be at the edge, rather than close to the middle of the range of runs (Figure R1 -- Uploaded as PDF with this review).

I think what is happening here is that the dark gray lines are plotted first, then the medium grey, then the light grey. This means that the dark lines are concealed by the medium grey lines. This leads the reader to believe the range of values is narrower than the mean value. Such a problem could possibly be solved (I think) if the plotting function randomized the order in which it plotted the different line colors. Alternatively, my suggestion might instead make everything even harder to read. I think readers can figure out what is actually going on as written, so it's fine to leave it if my suggestion doesn't help.

Thank you for this suggestion – we have made this change and agree it helps with clarity. We also now include a supplemental figure (*Supplement* Figure S33) where each run type is plotted separately so that you can clearly see the associated gray lines.

All the best,

Jacob

Figure R1. An example of how the mean flux profiles don't seem to be averages of the *shown* model run flux profiles. Similar discrepancies can be seen in most other panels.

REVIEWER #2 (Remarks to the Author):

Generic comments on Trang (to accompany the 'post-it' comments on the pdf)

The manuscript is much improved – and given its ambition (which I applaud) many clarifications are needed throughout the text. In particular for model parameterisations based on sparse observations or very limited experimental evidence. There are some very sweeping statements that need to be carefully qualified. I provide some examples of these below. Despite pointing out in the initial review that POM flux curves are driven by other biota than microbes – the authors persist with direct comparisons with the POM flux. This is a useful comparison but only if it is couched in the right manner, since a comparison will reveal how much influence – the model as parameterised – can have on shaping the POM flux profile and hence how influential the microbes might be in setting this cumulative flux.

Thank you again for your incredibly helpful comments! We have revised the text to provide the appropriate caveats and correct framing (*described in detail below*).

This relates to the following lines

Line 56 - but also flux feeders consuming particles

Thank you – we have added in a reference to flux feeding.

Line 74 - surely its how the microbes contribute to setting the shape of the flux profile - and does their contribution vary with depth - and why

Thank you for pointing this out, we have made this suggested text change to clarify that we are referring to how the microbes contribute to setting the shape of the flux profile (lines 54 and 78). While we are not able to directly assess the relative contribution of microbial POC degradation and zooplankton grazing, we do assess how microbial consumption varies with depth in the model (lines 174-177, *Supplement S3*, and *Supplement Figs. S10, S11, and S29*). We believe that the analyses and hypotheses presented in this study will help future studies better quantify the contribution of microbial degradation in situ and how it varies spatially (both regionally and with depth) and temporally. We highlight this point on lines 230-236, 240-256, and 281-287.

The authors also use a sinking rate linked to particle size when many studies – such as McDonnell et al. (2010) reveal that specific gravity not size is the main driver.

Thank you – we have edited the text (line 118 and *Supplement S1*) to indicate that it is changes in specific gravity of the particles that impacts sinking speed. Following previous studies^{1,2}, we model our particles using a fractal approximation such that the specific gravity of the particles changes with size (*Supplement S1, Supplement eqs. s13-15*). Large particles have lower specific gravity and smaller particles have higher specific gravity. As the particles in the model are consumed, the radius shrinks, and the specific gravity increases. As a result, the sinking speed is also impacted.

Line 108 - why would sinking speed decrease - surely its set by the integrated specific gravity of all of the different particle components - no relationship between size and sinking rate - its driven by specific gravity

if low specific gravity components of the particle (highly hydrated materials) are initially removed then particles

Thank you for this point. We have changed to using specific gravity rather than density in the text. In the expanded methods (*Supplement S1*), we discuss our simplifying assumption following previous studies such as³ that the reference density doesn't change and therefore changes in specific gravity scale linearly with changes in particle density. Multiple ways to define the relationship between particle size and particle sinking speed have been proposed in the literature (e.g.,^{2,4}). Based on *in situ* observations and previous modelling studies^{1,2,5,6}, we chose to represent particles across a spectrum from dense, carbon rich small particles to larger, fluffier, less carbon rich aggregates^{1-3,5,6}. By relating size to specific gravity of the particle, we are able to calculate sinking speed using Stokes Law⁶ (*Supplement eq. s15*). We confirmed that our results were not sensitive to our choice of the relationship between size and specific gravity, or size and carbon content of the particles (*Supplement S2, Supplement Fig. S20b and S21*).

Fig S20. Comparison of different particle density and particle sinking relationships. The relationship between particle radius r_i and particle carbon concentration $C_{poc,i}$ (a), and sinking rate ω_i (b) are shown for the formulation used in this paper (after Omand *et al* 2020) and the fitted equations of Alldredge *et al* (1998) and Bianchi *et al* (2018). Despite differences in carbon content and sinking speeds, our mechanistic model based on Omand *et al* 2020 shows quantitatively similar results to Alldredge *et al* (1998) (see *Supplement S2 and Supplement Fig. S21*).

Also the authors need to clarify where in the water column does colonisation take place – shallower than 50 m? at 50 or 100 m? the particle export initiation depth they have selected, throughout the water column?

Thank you for this suggestion. We have added text clarifying how colonization occurs in the model (lines 115-117). The particle is initialized with a stochastically assigned initial microbial biomass. This represents the cumulative impact of particle colonization that occurred above the

model ‘formation depth’ where we start simulating microbial dynamics. The particle is then continually colonized as it sinks through the water column by the free-living microbial pool.

Regarding your note in the pdf version of the text on the model colonization times: The colonization time (residence time) of particles in the model ranges from a few days (~5 days) to several months (~200 days) depending on the biotic and abiotic parameters with most particles lasting for a few weeks (~30 days, lognormally distributed) below the export depth, consistent with observations^{7,8}. We have added this analysis to the main text (line 146-148) and the *Supplement* (“*Supplement S3: Model dynamics*” section “*Variable remineralization rates*”).

130 ‘When population density at particle formation is high’

so at 50 m or 100 m - clarify

Thank you – we hope our clarification on how colonization occurs in the model has helped to clarify this. By “When population density at particle formation is high”, we mean that if a lot of colonization occurs in the upper water column (i.e., above the ‘export depth’ whether that is 50, 75 or 100 m), then we see higher rates of POC flux attenuation (fewer struggling particle-associated communities and more POC consumption by the microbial community).

can particles also form deeper in the water column??

For simplicity, for each model run, we use a single export depth (between 50-100 m depending on the simulation) and we do not allow additional particle formation within the water column. The impact of aggregation and disaggregation have previously been shown to be important⁹⁻¹¹. However, including these dynamics in our model is beyond the scope of this study. We now clarify that we make the simplifying assumption that no new particle formation occurs within the water column in the model (line 110-111, and lines 305-309).

We did conduct a sensitivity study to test the impact of varying the particle formation depth in the model. This analysis shows that variable formation depths create substantial variability in the POC flux in the upper 500 m. In particular, releasing particles at shallower depths allows faster POC surface consumption (*Supplement* Fig. S19) due to higher encounter rate and less influence of temperature. However, we see convergence of the model results below 500 m (*Supplement* Fig. S19). These results suggest that variable formation depth is an important factor in setting the POM flux in the upper ocean (<500 m) and that our model can provide some insight into this process. For example, for a given site, the model can provide a hypothesis as to the depth(s) of formation that best matches the observed POC profiles.

The demarcation regarding free living and attached bacteria is never made explicitly – causing issues that need clarity – such as

Thank you. We have clarified this in the text (lines 94, and 334-338).

Line 139

“from the free-living microbial pool as the particle sinks if the encounter rates”

how is this set - it typically decreases rapidly with depth?

We assume that the abundance of motile POC degrading bacteria decreases by approximately an order of magnitude between the export depth and 2,000 m following the observed decrease in abundance of the total free-living bacterial pool¹². The non-particle associated or free-living community of bacteria in group i (F_i) is one of the unconstrained parameters in our model and so we vary F_i over quite a wide range. We have edited the text to highlight that F_i decreases exponentially with depth, the uncertainty in this parameter, and the need for future studies to better constrain this term in the model (lines 114-115, 336-338, and *Supplement S1*).

See above generic comments on poorly resolved terms – sparse observations and / or very limited (if at all) experimental evidence. These ‘weak’ terms need to be flagged in the results section to give the reader some idea of the caveats associated with using microbial ecophysiology for subsurface waters.

Thank you for highlighting the need to further emphasize the poorly constrained terms in our model. We have added text to clarify this (lines 99-102, and 283-287). We also have made sure to highlight uncertain model parameters and how we define realistic ranges for the model in the extended methods (*Supplement S1*)

lines 161-163 bacteria growth effic at depth is something that is poorly resolved
as this is a major conclusion

We completely agree with Dr. Boyd and have added text to emphasize that better constraints are needed on this (and other) important microbial terms (lines 99-102, and 283-287).

varied temperature profiles. Temperature does play an important role in the loss processes
described above as decreased growth rates due to temperature limitation makes it more
difficult

for microbial populations to overcome loss processes and successfully colonize particles.
However, our results suggest that temperature limitation is not the primary driver of the
observed
dynamics

of growth rate - depends on how it is parameterised - are there any data in the literature on
thermal performance curves? – conduct some sensitivity analysis.

Thank you for highlighting this point! We use two different, commonly used, relationships between temperature and growth rate^{3,13}. However, we agree with the importance of validating these curves with experimental data. We now include a figure (*Supplement Fig. S34*, referenced on line 123) comparing the modeled temperature-growth rate relationship to published thermal performance curves for marine heterotrophs and to two previously unpublished thermal performance curves for a chitin degrading marine *Vibrio*. We have conducted a suite of sensitivity analyses with different thermal performance curves and with different water-column temperature profiles (described in *Supplement S3*, *Supplement Figs. S4, S7, S16, S27, S18, S31, S34*). These sensitivity studies indicate that temperature plays a secondary role and that other processes, such as depth of particle formation, particle lability, encounter rate, and particle size spectrum, are more important for setting the rate of POC flux attenuation.

line 192 The explicit representation of micro-scale dynamics on particles in our model
generates water; 193 column-scale estimates of POM fluxes that are consistent with
observations^{1,3}

but the POM flux profiles are set by more than microbes - see my comments in the original submission

We have edited the text to clarify our point that the microbial impact on POC flux attenuation can vary significantly and may play an important role in contributing to the shape of the POC flux profile, in addition to other processes such as grazing and aggregation/disaggregation.

REFERENCES

1. Jackson, G. A. Comparing observed changes in particle size spectra with those predicted using coagulation theory. *Deep Sea Res. Part II Top. Stud. Oceanogr.* **42**, 159–184 (1995).
2. Alldredge, A. The carbon, nitrogen and mass content of marine snow as a function of aggregate size. *Deep Sea Res. Part Oceanogr. Res. Pap.* **45**, 529–541 (1998).
3. Cram, J. A. *et al.* The Role of Particle Size, Ballast, Temperature, and Oxygen in the Sinking Flux to the Deep Sea. *Glob. Biogeochem. Cycles* **32**, 858–876 (2018).
4. Bianchi, D., Weber, T. S., Kiko, R. & Deutsch, C. Global niche of marine anaerobic metabolisms expanded by particle microenvironments. *Nat. Geosci.* **11**, 263–268 (2018).
5. McDonnell, A. M. P. & Buesseler, K. O. Variability in the average sinking velocity of marine particles. *Limnol. Oceanogr.* **55**, 2085–2096 (2010).
6. Omand, M. M., Govindarajan, R., He, J. & Mahadevan, A. Sinking flux of particulate organic matter in the oceans: Sensitivity to particle characteristics. *Sci. Rep.* **10**, 5582 (2020).
7. Boyd, P. W. *et al.* Transformations of biogenic particulates from the pelagic to the deep ocean realm. *Deep Sea Res. Part II Top. Stud. Oceanogr.* **46**, 2761–2792 (1999).
8. Schmidt, S., Chou, L. & Hall, I. R. Particle residence times in surface waters over the north-western Iberian Margin: comparison of pre-upwelling and winter periods. *J. Mar. Syst.* **32**, 3–11 (2002).
9. Burd, A. B. & Jackson, G. A. Particle Aggregation. *Annu. Rev. Mar. Sci.* **1**, 65–90 (2009).

10. Buesseler, K. O. & Boyd, P. W. Shedding light on processes that control particle export and flux attenuation in the twilight zone of the open ocean. *Limnol. Oceanogr.* **54**, 1210–1232 (2009).
11. Jackson, G. A. & Burd, A. B. A model for the distribution of particle flux in the mid-water column controlled by subsurface biotic interactions. *Deep Sea Res. Part II Top. Stud. Oceanogr.* **49**, 193–217 (2001).
12. Sunagawa, S. *et al.* Tara Oceans: towards global ocean ecosystems biology. *Nat. Rev. Microbiol.* **18**, 428–445 (2020).
13. Eppley, R. W. Temperature and phytoplankton growth in the sea. *Fish. Bull.* **70**, 1063–1085 (1972).

REVIEWER #3 (Remarks to the Author):

I appreciate the detailed response to the review comments, including the additional simulations. Some of the potential shortcomings or unknown effects (e.g., temperature, different paths of OM degradation) have been addressed. Other missing pieces are clearly identified, most importantly the lack of accounting for zooplankton; particle aggregation/disaggregation. The latter remains a significant question mark, albeit one that goes well beyond the scope of this paper. I consider this manuscript a very interesting addition to the discussion on the controls of POM fluxes, as it connects processes at the particle scale with POM fluxes and highlights the role of microbial dynamics.

Thank you again for your constructive review!

typos, minor comments

Line 43 - should be “contribution”

Thank you – we have made this change.

Caption figure 3: define UVP

Thank you – we have made this change.

Supplement 3, line 337: consider expanding, and point to the relevant equations you use to derive eq. s22-s24

Thank you for this suggestion. We have added a new supplemental section (*Supplement S4*) in which we include a full derivation of eq s22-s24 from the previous submission. We also include a new figure to accompany *Supplement S4* illustrating the dynamics of density-dependent growth on the surface of particle organic carbon.

Supplement 3, line 442 - s23 instead of s26

Thank you – we have made this change.

1 **Impact of microbial activities on the ocean particulate carbon flux**

2

[revised manuscript text omitted]

Deleted: 1b
 Deleted: .
 Deleted: .

Deleted: 1b

Deleted: function of microbial dynamics (e.g. growth rate, uptake kinetics, and yield) and particle chemical and physical properties (e.g. particle size and monomer diffusivity) (Supplement S3).

Deleted: This

Deleted: P

Deleted: p

Deleted: size

Deleted: microbial

Deleted: s

Deleted: cc

Formatted: Font: Italic

Formatted: Font: Italic

Deleted: (the ‘Allee effect’⁵⁵) arises in our model due to the interplay between diffusive nutrient loss, saturating (Monod) growth kinetics, and mortality (Supplement Figs. S6-S9).

Deleted: POM

Deleted: 2b

Deleted: pom

Deleted: microbial

Deleted:

Deleted: set

Deleted: 2

Deleted: POM

Deleted: 2

Deleted: 2

Deleted: 2d

Formatted: Font: Italic

Deleted: POM

Deleted: LMWOM

Deleted: POM

[revised manuscript text omitted]

Deleted: POM

Deleted: that are consistent with observations^{1,3} (*Fig. 34*).

Deleted: POM

Deleted: POM

Deleted: POM

Deleted:

Moved (insertion) [1]

Deleted: Our model captures the mean, range, and distribution of observed POM flux profiles from across the global oceans (n=897)³ (*Fig. 34*) suggesting that variability in microbial dynamics can play a significant role in variable POM fluxes. Higher b values are associated with higher abundance of small particles (flatter particle size spectra) and higher lability. Thus, oligotrophic regions with flatter particle size spectra are predicted to coincide with higher b values, consistent with previous studies^{5-9,16}.

Deleted: As expected, differences in particle size plays a key role in the efficiency of carbon transfer as larger particles sink substantially faster (*Fig. 2 3 and Supplement Fig. S20*). However, we show that the particle-associated microbial community dynamics can alter the POM flux to the same extent as a change in particle size (*Fig. 2d3d, Supplement Fig. S22*). These model dynamics (from rapidly growing populations to those struggling to survive) emerge as a result of stochastic interactions between biological, chemical, and physical controls on microbial growth.

Moved up [1]: These model dynamics (from rapidly growing populations to those struggling to survive) emerge as a result of stochastic interactions between biological, chemical, and physical controls on microbial growth.

Deleted: POM

Deleted: with

Deleted: However, because the persistence of POMPOC in the deep ocean (>1000 m) is generated by

Deleted: POM

Deleted: to depth

Deleted: 3

Deleted: POM

[revised manuscript text omitted]

Deleted: POM

Deleted: POM

Deleted: POM

Deleted: POM

Deleted: POM

Deleted: POM

Deleted: is temporally and spatially variable as a function of microbial growth dynamics

Deleted: POM

Deleted: predictions can

Deleted: , which

Deleted: POM

Deleted: also

Deleted: matter

Deleted: POM

Deleted: particle-associated

Deleted: matter

include enzymatic degradation of polymer into low molecular weight organic matter (C_{lmwoc}),
 density dependent growth of the particle-associated microbial community (B_i), and the attachment
 ($E_{i,z}$) and detachment (L_i) of heterotrophic microbes to/from the particles. This model can be
 coupled to a full ecosystem model such that the generation of each particle type can be calculated
 prognostically. However, here we focus on the degradation of POC below the export depth and so
 simply include a source term to represent net particle formation above the export depth (default
 100 m. see Supplement S2 for simulations with alternative formation depths). For simplicity, we
 also do not allow for aggregation or disaggregation to occur within the water column. An extended
 model description is provided in Supplement S3.

- Deleted: m
- Deleted: heterotrophic
- Deleted: microbes
- Deleted: POM
- Deleted: at 100 m
- Deleted: was

491 The change in the carbon content of particle i ($C_{poc,i}$, mmol C_{poc} particle⁻¹) over time is defined
 as:

$$493 \frac{dC_{poc,i}}{dt} = -\beta_i B_i \quad \text{eq. 1}$$

where β_i (mmol C_{poc} mmol C_{cell} ⁻¹ day⁻¹) represents the polymer degradation rate of $C_{poc,i}$ by
 microbial population B_i (mmol C_{cell} particle⁻¹) similar to⁶⁰. Specifically, β_i captures differences
 in 'lability' of particles, which is a function the organic carbon itself, the microbial enzymes
 specific to group B_i , and production rate of those enzymes by B_i . When the particle is fully
 consumed ($C_{poc}=0$), the particle-associated microbial community detach and so consumption
 stops.

- Deleted: m
- Deleted: pom
- Deleted: m
- Deleted: pom
- Deleted: m
- Deleted: matter
- Deleted: pom

The enzymatic degradation of POC results in the production of low molecular weight organic
 matter (LMWOM) (C_{lmwom} , mmol C m⁻³ particle⁻¹) which supports microbial growth. We assume
 that there is no loss of carbon during the enzymatic cleavage from POC to LMWOM such that 1
 504 mmol C_{poc} degraded = 1 mmol C_{lmwom} produced. There is however diffusive loss of LMWOM
 away from the particle as described in eq. 2. Specifically, the LMWOM concentration is calculated
 assuming steady state dynamics as:

$$507 C_{lmwom} = (\beta_i B_i - \frac{\mu_i B_i}{y_{lmwom}}) / d_{loss} \quad \text{eq. 2}$$

where μ_i is the growth rate of microbes B_i on the particle (day⁻¹), y_{lmwom} is aerobic microbial
 growth efficiency (mmol C_{cell} mmol C_{lmwom} ⁻¹), and d_{loss} is the diffusive loss rate of LMWOM (m³
 510 day⁻¹) (Supplement S1 eq. s3 and Supplement S4 for full calculation).

- Deleted: POM
- Deleted: POM
- Deleted: pom
- Deleted: m
- Deleted: -
- Deleted: t
- Deleted: LMWOM
- Deleted: LMWOM

511 Microbial dynamics on each particle are defined as:

$$513 \frac{dB_i}{dt} = B_i(\mu_i - L_i - m_{lin,i}) + E_{i,z} \quad \text{eq. 3}$$

where $m_{lin,i}$ is microbial mortality rate (day⁻¹). L_i is the detachment rate (day⁻¹). The microbial
 encounter rate ($E_{i,z}$, mmol C_{cell} day⁻¹ particle⁻¹) represents the rate of colonization of the particle
 by the free-living microbial pool. $E_{i,z}$ varies with depth based on particle size, sinking speed, and
 the abundance of free-living and motile microbes in group i in the water column^{52,65} (eq. s11 in
 Supplement S1). The free-living abundance is assumed to decrease exponentially with depth⁶⁵.
 Microbial growth rate, μ_i (day⁻¹) is dependent on the LMWOM concentration at the particle
 surface and is represented with the Monod equation:

$$521 \mu_i = V_{max,i} \frac{C_{lmwom}}{C_{lmwom} + k_{m,i}} \gamma_{T,z} \quad \text{eq. 4}$$

where $V_{max,i}$, $k_{m,i}$, and $\gamma_{T,z}$ represent maximum LMWOM uptake rate, half saturation of LMWOM
 uptake of microbial group B_i , and temperature limitation at depth z (eq. s11 in Supplement S1),

- Deleted: B_i
- Deleted:
- Deleted: in the water column

respectively. Model parameter values are given in *Supplement* Table S1 and sensitivity tests are
described in *Supplement* S2.

The total **POC** flux at a certain depth z , $F_{poc,z}$ (mmol C_{poc} m^{-2} day^{-1}), is calculated as the **sum** of
the vertical fluxes of each individual particle as they sink through the water column, where $F_{poc,z}$
is:

$$554 \quad F_{poc,z} = \sum C_{poc,i,z} N_{i,z} \omega_{i,z} \quad \text{eq. 5}$$

where $N_{i,z}$ is the number of particle type i per m^3 water column at depth z .

To test the impact of particle associated microbial dynamics on the **POC** vertical flux through the
water column, we perform a set of stochastic simulations in which parameters are randomly chosen
from within a reasonable range. Here we simulate a single $2.24 \text{ m} \times 2.24 \text{ m}$ column initialized
between 50 and 100 m with an observed particle size distribution yielding a total flux of 1.5 mol
$m^{-2} \text{ yr}^{-1}$ ^{1,5,13}. Specifically, we simulate 2,256 particles (particle size spectra with power law
exponent $s = -2$), 69,000 particles ($s = -3$), or 763,615 particles ($s = -4$) ranging from $50 \mu\text{m}$ to 0.4
563 cm in radius as they fall through the water column. For each particle, the following model
parameters are stochastically assigned within a reasonable range (*Supplement* Table S1) using a
uniform distribution, except for lability for which a log-normal distribution is used: maximum
growth rate V_{max} (1.2 or 7.2 day^{-1}), particle lability β ($10 - 1,000 \text{ mmol } C_{poc} \text{ mmol } C_{cell}^{-1} \text{ day}^{-1}$),
initial cell density ($400 - 2,800 \text{ cell mm}^{-2}$), and density of free-living community of microbes B_i
(F_i) (10 to 285 cell mm^{-3}). The stochastic simulations are conducted 1,000 times for each particle
size distribution. Simulations are run for 600 days which is sufficient for all particles to be fully
consumed or exported to $> 4,000 \text{ m}$. The attenuation exponent b for the modeled **POC** flux is
calculated using a least square fit of the power law function.

Deleted: POM

Deleted: m

Deleted: pom

Deleted: accumulation

Deleted: m

Deleted: m

Deleted: m

Deleted: POM

Deleted: pom

Deleted: POM

**Acknowledgement**

This work was supported by a grant from the Simons Foundation (542387 to NML, 542395 to RS
and OC, 542389 to TH). We thank N. Norris and E. Lee for assistance with model development.
We thank our reviewers (Dr. J. Cram, Dr P. Boyd, and an anonymous reviewer) for their
constructive feedback.

Deleted: W

**Author contributions**

589 T.T.H.N and N.M.L designed the study, developed the full model, and conducted the numerical
analysis; TC and TH developed and analyzed the mathematical model in Supplement S4; KA and
TC performed measurements of bacterial parameters for *Vibrio* sp. 1A01; E.J.Z, O.X.C, A.E, J.S,
U.A., F.J.P, T.H, and R.S contributed data and to model development; all authors contributed to
the writing.

**Competing interests**

The authors declare no competing interests.

**Data availability**

No new data is presented in this paper.

**Code availability**

The model code will be deposited on GitHub (<https://github.com/LevineLab/POCmodel>) and is
currently available at <https://dornsife.usc.edu/labs/levine/levinelabcode/>.

Deleted: POM

**Figure captions**

**Figure 1. Micro-scale model dynamics.** (a) Illustration showing micro-scale model dynamics
occurring on sinking particles. Primary degraders (red microbes) convert polymeric organic matter
(dark blue sphere) into low molecular weight organic matter (LMWOM, light blue) using
extracellular enzymes (yellow). The particle-associated community experiences loss due to
mortality (gray) and detachment (purple). (b) Illustration of water column model dynamics with
an emphasis on a single particle (blue sphere) falling through the water column. Each particle is
randomly assigned with an initial radius, lability, and set of biological parameter values at the
depth of formation (see *Methods*). The particle-associated microbial dynamics then evolve
prognostically for each particle as it sinks through the water column and is consumed by microbial
activity. The total POC flux throughout the water column is obtained by summing across all
sinking particles.

**Figure 2. Density-dependent growth validation.** The model captures observed density-
dependent growth of particle-associated communities. Fold change in microbial population
biomass after 10 hours of growth is shown for model simulations (red asterisks) and experimental
data (open circles, Ebrahimi *et al* (2019)).

**Figure 3. Particle-associated microbial growth dynamics.** Example degradation dynamics are
shown for rapidly growing populations (curves E, F, G), 'rescued' populations (C, D), and
'struggling' populations (A, B). Parameter values are given in *Supplement Fig. S12*. For each
simulation, the change in microbial density on the particle surface through time (a), percentage
changes in the particle radius over time (b) and depth (c) are shown. To quantify the impact of
microbial growth dynamics on POC persistence at depth, the fraction of particles within a given
lability class and radius at formation that persist at 1,000 m is estimated (d). The colors and values
in each grid correspond to the percentage of model parameter combinations for a given lability and
initial radius (n=72) that persist past 1,000 m depth (see *Supplement Methods S1*).

**Figure 4. Microbial contribution to large-scale POC fluxes.** Shifts in POC fluxes over depth
are shown as a function of varying particle and microbial dynamics. Each gray line is the integrated
flux over 2,256 to 763,615 stochastically initialized particles (depending on particle size spectra).
For each parameter set, 1,000 stochastic simulations were conducted (1,000 gray lines of different
shades) and the average is shown as the thick colored line. The data from Martin *et al* (1987) (in
open circles) are also shown. POC transfer efficiency decreases (larger attenuation exponent b)
with more negative particle size spectra power law exponent s (a), higher mean particle lability
(β_{avg} in $\text{mmol C}_{\text{POC}} \text{mmol C}_{\text{cell}}^{-1} \text{day}^{-1}$) (b), higher maximal growth rate (μ_{max} in day^{-1}) (c), and
higher particle-microbe encounter rate (Enc) (d). The distribution of attenuation exponent (b) for
all 10,000 stochastic simulations from panels a-d (gray bars) are compared against observed
attenuation coefficients from 897 global Underwater Video Profiler (UVP) samples compiled by
Guidi *et al* (2015) (open bars) (e).

Formatted: Font: 12 pt

Formatted: Font: 12 pt

Formatted: Font: 12 pt

Formatted: Font: 12 pt

Formatted: Font: 12 pt

Formatted: Font: 12 pt

645

646 **Reference**

[revised manuscript text omitted]

- 842

REVIEWERS' COMMENTS

Reviewer #2 (Remarks to the Author):

I thank the authors for their patience in responding to my many comments. They have done an admirable and thorough job of responding in detail to my comments. The results and discussion read very clearly now. I have added a few suggested modifications in the attached pdf to improve the flow in the Introduction and a few suggested changes to the abstract. i look forward to see this insightful study published.

Philip Boyd